

# Global modeling of primary biological particle concentrations with the EMAC chemistry-climate model

Meryem Tanarhte[1,2], Sara Bacer[1], Susannah M. Burrows[3], J. Alex Huffman[4], Kyle M. Pierce[4], Andrea Pozzer[1], Roland Sarda-Estève[5], Nicole J. Savage[4*], Jos Lelieveld[1,6]

[1]*Max Planck Institute for Chemistry, Department of Air Chemistry, Mainz, Germany*

[2]*University Hassan II-Casablanca, Faculté des Sciences et Techniques, Mohammedia, Morocco*

[3]*Atmospheric Science and Global Change Division, Pacific Northwest National Laboratory, Richland, WA, USA*

[4]*University of Denver, Department of Chemistry and Biochemistry,, Denver, USA*

[5]*Laboratoire des Sciences du Climat et de l'Environnement, CEA/CNRS-UVSQ, 91191, Gif/Yvette, France*

[6]*The Cyprus Institute, Nicosia, Cyprus*

*\*Now at Aerosol Devices, Inc.*

*Correspondence to*: Meryem Tanarhte (meryem.tanarhte@mpic.de)

**Abstract.** Primary biological aerosol particles (PBAPs) may impact human health and aerosol-climate interactions. The role

of PBAPs in the earth system is associated with large uncertainties, related to source estimates and atmospheric transport. We used a chemistry-climate model to simulate PBAPs in the atmosphere including bacteria, fungal spores and pollen. Three fungal spore emission parameterizations have been evaluated against an updated set of spore counts synthesized from observations reported in the literature. The comparison indicates an optimal fit for the emission parameterization proposed by (Heald and Spracklen, 2009), although the model significantly over-predicts PBAP concentrations in some locations.

Additional evaluation was performed by comparing our combined bacteria and fungal spore simulations to a global dataset of fluorescent biological aerosol particle (FBAP) concentrations. The model predicts the sum total of measured PBAP concentrations relatively well, with an over- or under-prediction of less than a factor of 2 compared to FBAP. The ratio of bacteria to fungal spores reflects a greater difference, however, and the simulated bacteria concentrations outnumber the simulated fungal spore concentrations in almost all locations. Further, the modeled fungal spore results under-predict the

FBAP concentrations, which are used here as a rough proxy for spores. Uncertainties related to technical aspects of the FBAP and direct-counting spore measurements challenge the ability to further refine quantitative comparison on this scale. We estimate that the global PBAPs mass concentration (apart from desert dust and sea salt aerosols), i.e. of fungal spores and pollen, amounts to 19% and 52% of the total aerosol mass, respectively.



## 1 Introduction

Primary biological aerosol particles (PBAPs) are diverse and include bacteria, fungal spores, viruses, pollen as well as fragments of other organisms. Their size and source characteristics are different throughout the globe. This influences their residence times and spatial distributions. The increasing interest in PBAP is related to the effects they may have on

agricultural crops, human health and atmospheric chemistry (Deguillaume et al., 2008). Airborne bioparticles may influence climate by acting as ice nuclei in mixed-phase clouds. The abundance of atmospheric ice nuclei (IN) can influence cloud development, impacting cloud radiative properties and the location and timing of rain formation (Bangert et al., 2012; DeMott et al., 2010; French et al., 2018; Prenni et al., 2007). PBAPs are likely to be important contributors to ice nuclei abundance under clean (or near-pristine) conditions in remote locations, for example over the Amazon rain forest (Pöschl et

al., 2010; Prenni et al., 2009). PBAPs have been shown to make an important contribution to the atmospheric aerosol mass in some locations (Bauer et al., 2002a; Després et al., 2012; Elbert et al., 2007). Since they widely range widely in size from approximately 0.01 to 100 μm, the time suspended in the air can vary from minutes to days (Després et al., 2012; Fröhlich-Nowoisky et al., 2016; Pöschl, 2005). Microorganisms have also been shown to be transported over long distances with desert dust from Asia and Africa (Griffin et al., 2007; Prospero et al., 2005; Smith et al., 2013), potentially contributing to

the global transport of genetic information.

PBAPs have been identified and characterized using a large range of methods, including traditional methods such as microscopic analysis and cultivation methods, and modern methods using molecular techniques (Caruana, 2011; Després et al., 2012; Griffiths and Decosemo, 1994). However, continuous measurements of PBAPs have been limited, and therefore actual abundances, properties, as well as the origin of PBAPs and their components are still poorly quantified and

understood (Burrows et al., 2009a; Burrows et al., 2009b). Fungal spores are the most abundant and the most genetically diverse PBAPs in the atmosphere (Lacey, 1981). Fungal spores are also of critical importance because many species can induce considerable economic losses, acting as plant pathogens or triggering respiratory diseases and allergenic processes in humans (Reinmuth-Selzle et al., 2017). Inhalation of spores in significant quantities causes various respiratory diseases such as allergic rhinitis, asthma, and other allergic reactions (Burge and Rogers, 2000; Bush and Portnoy, 2001). More than 100

species have been shown to contribute to respiratory disorders (Green et al., 2005).

Estimates of the total global emissions of fungal spores emitted into the atmosphere diverge greatly across the literature, varying from 8 Tg yr$^{-1}$ (Sesartic and Dallafior, 2011) to 186 Tg yr$^{-1}$ (Jacobson and Streets, 2009). Fungal spores contribute up to ~ 45% of the coarse particulate matter over the tropical rainforest (Després et al., 2012) and their number and mass concentrations are typically about $10^3$ to $10^4$ m$^{-3}$ and ~1 μm m$^{-3}$, respectively (Fröhlich-Nowoisky et al., 2016). The number

and composition of airborne fungal spores depends on complex interactions between biological and environmental factors, including the climate and local ecological systems (Grinn-Gofron and Bosiacka, 2015). Meteorological factors are known to influence their production, release and transport, which is contingent to geographical areas, vary seasonally, and interactions



with PBAP can also depend on the emitting species involved (Elbert et al., 2007; Fröhlich-Nowoisky et al., 2009; Hirst, 1953; Levetin and Dorsey, 2006; Li and Kendrick, 1995; Oliveira et al., 2009).

Global and regional models have been used to evaluate PBAP emissions, transport and their impact on the hydrological cycle by acting as CCN and IN (Burrows et al., 2009a; Heald and Spracklen, 2009; Hoose et al., 2010; Hummel et al., 2015;

Jacobson and Streets, 2009; Sesartic and Dallafior, 2011; Spracklen and Heald, 2014). These models require the credible representation of the emissions and particle properties influencing transport and removal from the atmosphere. Large uncertainties in the number concentrations remain, especially due to inherent uncertainties in the emission estimates of PBAPs.

The goal of this study is to evaluate three fungal spore emission parameterizations available in the literature and compare

their number concentrations, simulated with a global model, to an updated set of observations synthesized from the literature. We use a chemistry-climate model to simulate the total PBAPs present in the atmosphere including bacteria, fungal spores and pollen. We compare our simulated PBAP concentrations to a global dataset of fluorescent biological aerosol particles (FBAP) concentrations. These measurements have been performed with real-time techniques that detect the fluorescence signal through UV excitation of fluorophores commonly present biological materials (e.g., fungal spores,

bacteria, and leaf fragments).

## 2. Materials and methods

### 2.1. Model description

The global ECHAM5/MESSy Atmospheric Chemistry – Climate (EMAC) model (ECHAM version 5.3.01, MESSy version 2.5.2 (Jöckel et al., 2005; Jöckel et al., 2016) was used to simulate the emissions and transport of biological particles. The primary biological aerosol particles have been included with the emission parameterizations described below.

Removal processes of particles simulated by the model include sedimentation, dry deposition, impaction scavenging, and

nucleation scavenging by liquid, mixed-phase, and ice clouds. PBAP dry and wet depositions are treated as described for other aerosol species (see (Burrows et al., 2009a; Pozzer et al., 2012; Pringle et al., 2010a)) and references therein). We simulated the transport of aerosol tracers of different sizes, as described in more detail below. All particle classes are treated as having a lognormal distribution with modal-scale parameter $\sigma = 1$ ($\sigma = 1.4$ for pollen) and with a density of 1 g cm$^{-3}$. We assume that all particles can become activated as cloud condensation nuclei when calculating particle removal processes by

wet deposition, as described in (Burrows et al., 2009a). All PBAPs are transported as passive tracers, i.e., their concentrations are influenced by model processes such as dry deposition and scavenging by clouds and rain, but do not interact with radiation or change cloud microphysical properties. The sedimentation and dry deposition of the particles are treated as described in (Kerkweg et al., 2006). Wet deposition of the particles is described in (Tost et al., 2006).





For the present study, we applied EMAC in the T63L31 resolution; with a spherical truncation of T63 (corresponding to a grid of approximately 1.9° x 1.9 ° in latitude and longitude, or approximately 140 **x** 210 km at middle latitudes), with 31 vertical hybrid pressure levels up to 10hPa. The model was run for five consecutive years without meteorological nudging from the year 2000 until 2004. AMIP-II monthly sea surface temperatures were used to provide boundary conditions for the

atmospheric circulation, available for the period since satellite observations are available (1979). Climatological averages for PBAP distribution for the last four years of the simulation were used after a 1-year spin-up period. We emphasize that the simulation results represent a climatology rather than specific weather conditions under which some PBAB samples may have been collected, hence we expect mean number concentrations and distributions to be represented by the model rather than distinct measurement data.

## 2.2. PBAP emissions

### 2.2.1. Bacteria

Bacterial emission fluxes are calculated using the best-estimate values from (Burrows et al., 2009a) for different ecosystems, which were optimized toward overall agreement with best-estimates of observation-based near-surface number concentrations. We used the MODIS International Global Biosphere Program (IGBP) global land cover classification to determine the spatial distribution of 18 different ecosystems. We lumped the categories defined in the MODIS classifications to match similar sets of lumped ecosystems used by (Burrows et al., 2009a) (i.e., derived from the Olson ecosystem types),

with the exception of the "urban" ecosystem, which is only present in MODIS data.  We used a geometric mean diameter for bacteria of 4 μm for continental sources (forests, shrubs, grasslands, wetlands, savannahs and urban ecosystems) and 1.4 μm for marine sources. These choices are based on values reported for the count median diameter of bacteria-carrying particles, which may include bacteria borne by larger particles such as dust and leaf litter and/or clumps of bacteria (Shaffer and Lighthart, 1997; Tong and Lighthart, 2000, 1999; Wang et al., 2007). We note that modeled transport and removal processes

are not strongly dependent on the particle size, at least not in the lower μm size range, so that we do not consider the simplified size attribution of PBAPs to be a limiting factor in the representation of atmospheric processes (Haga et al., 2014; Kunkel et al., 2012).

### 2.2.2. Fungal spores

We compare three fungal spore emission parameterizations previously used in global and regional modeling studies. Firstly, fungal spore emission fluxes have been derived by (Heald and Spracklen, 2009) (HS hereafter)  from an empirically optimized scheme where emissions are linear functions of the LAI (Leaf Area Index) and the specific humidity q at the





surface. In order to match their emission estimates, (Hoose et al., 2010) (applied the following formulation to calculate the emission flux in m$^{-2}$ s$^{-1}$, assuming a mean spore diameter of 5 μm:

$$F_{H\&D} = 500 m^{-2}s^{-1} \times \frac{LAI}{5} \times \frac{q}{1.5 \times 10^{-2}kgkg^{-1}}$$

5      The second parameterization we tested uses the emission number fluxes of fungal spores calculated by (Sesartic and Dallafior, 2011) (SD hereafter) for five different ecosystems (defined by (Olson et al., 2001)). We use the best-estimate number fluxes weighted by the area fraction of the respective MODIS ecosystems in the gridbox $E_i$. Note again that similar ecosystems from MODIS data are lumped according to the corresponding Olson ecosystems defined by (Sesartic and Dallafior, 2011). The total emission flux for fungal spores is given as in m$^{-2}$ s$^{-1}$:

$$F_{S\&D} = 194 \, m^{-2}s^{-1} \times E_{tropicalforest} + 214 \, m^{-2}s^{-1} \times E_{forest} + 1203 \, m^{-2}s^{-1} \times E_{shrub} + 165 \, m^{-2}s^{-1} \times E_{grassland}$$
$$+ 2509 \times m^{-2}s^{-1} \times E_{crop}$$

The third parameterization, derived by (Hummel et al., 2015) (HU hereafter), is adapted to measurements of airborne fluorescent biological particles across northern Europe. Similar to the parameterization of (Heald and Spracklen, 2009), this recent parameterization depends on LAI and specific humidity, and is extended to include temperature T:

$$F_{FBAP} = b_1 \times (T - 275.82) + b_2 \times q \times LAI$$

where $F_{FBAP}$ the emission flux in m$^{-2}$ s$^{-1}$, b1 = 20.426 and b$_2$ =3.93 10$^4$, T is the surface temperature in K, q the specific
15    humidity in kg kg$^{-1}$ and LAI the leaf area index in m$^2$m$^{-2}$.

For each parameterization, the mean diameter was assigned according to the recommendation made for each in the three studies: 5 μm for HS, 3 μm for HU and 3 μm for SD. We reiterate that these size classifications are not expected to significantly influence the results.

**2.3. Data description**

20    **2.3.1. Spores counts**

We compare the fungal spore concentrations calculated by EMAC using the three emission parameterizations described above, to observations collected through literature review. (Sesartic and Dallafior, 2011) have reviewed more than 150 studies and found that only a relatively small number, 35 of these, reported total fungal spore concentration measurements,



excluding observations that employed cultivation of a subset of species (e.g., in a petri dish) and measurements that report only mass concentrations instead of spore counts. We updated this dataset with observations that meet the same criteria, mostly from studies published since 2011. Our updated review revealed that much of the relevant literature reports only concentrations of the genetic diversity of fungal taxa and not their total concentrations, which explains the scarcity of data

that can be used for model evaluation. The uncertainties related to these methods are discussed in detail in (Sesartic and Dallafior, 2011). The observations used for comparison with model results are listed in Table SI1. Modeled concentrations have been sampled from the output to match the period of observation for each location. Since we do not compute actual meteorology but rather climatological conditions, our model results do not represent instantaneous local processes, especially when they vary strongly on a small scale. However, we expect that time averaging limits such biases. We

differentiate the data by ecosystem using the MODIS categories and the description provided by the reference. Most of the observations that met our criteria for inclusion have been taken in urban areas.

### 2.3.2. FBAP observations

Over the last two decades real-time measurement techniques have provided opportunities to monitor airborne PBAP continuously at relatively high time-resolution. Techniques involving laser/light-induced fluorescence (LIF) have been particularly effective in rapidly providing information about PBAP in real time (e.g. (Fennelly et al., 2018; Huffman and Santarpia, 2017; Kaye et al., 2005; Pan et al., 2009; Saari et al., 2014; Sivaprakasam et al., 2009)). Among many available instruments, two commercially available LIF biosensors have been widely applied to ambient bioparticle monitoring and

have helped to reveal fine detail about atmospheric PBAP patterns not previously observed (Gabey et al., 2010; Huffman et al., 2013; Huffman et al., 2010; Perring et al., 2015; Schumacher et al., 2013). For example, the ultraviolet aerodynamic particle sizer (UV-APS; TSI, Inc.) and the wideband integrated bioaerosol sensor (WIBS; University of Hertfordshire or Droplet Measurement Technologies) both characterize biological particles in real-time based on the intensity of fluorescence emission observed from individual particles after pulsed excitation at wavelengths characteristic for common

biofluorophores (Foot et al., 2008; Hairston et al., 1997; Pöhlker et al., 2012). Despite the uncertainties related to this type of measurement (Huffman et al., 2012; Pöhlker et al., 2012; Savage et al., 2017), FBAP detected by the UV-APS or WIBS have been successfully used in some cases as a lower-limit for the atmospheric abundance of PBAPs in the super-micron (> 1 μm) size range (Huffman et al., 2010).

In this context, however, it is important to mention a few caveats implicit with the assumption linking FBAP to PBAP. First, real-time LIF instruments can only detect the physical properties of particles (i.e., fluorescence and size) and cannot directly determine whether a particle is of biological origin. By applying certain analytical strategies, however, a given ensemble of particles may be assigned as PBAP with varying degrees of certainty. In some cases weakly fluorescing biological particles can escape LIF detection (e.g. (Huffman et al., 2012)) and in other cases highly fluorescent particles of non-biological origin





can interfere with LIF detection to overestimate PBAP (e.g. (Gabey et al., 2013; Huffman et al., 2010; Saari et al., 2013; Savage et al., 2017)). Bioparticle size also plays an important role in LIF detection. For example, viruses are generally too small to be detected by LIF instruments, and almost all species of pollen are too large to be detected without fragmentation or instrument modification (O'Connor et al., 2011). Additionally, technical differences in instrument design, the choice of

detection channels, and operational parameters can have significant effects on the reported number concentration of FBAPs and the quality of their correlation with PBAP classes (Savage et al., 2017). Nevertheless, we use the FBAP numbers reported by UV-APS and WIBS instruments as a rough proxy for PBAP, comparing the observed FBAP numbers both with more direct PBAP measurements and with model outputs.

We show FBAP observations using UV-APS and the FL3 channel from the WIBS-3 and WIBS-4A instruments (Table 2). The majority of FBAP data shown were extracted from published reports without additional analysis or as tabulated by previous reviews (Fennelly et al., 2018; Saari et al., 2015; Yu et al., 2016). All original data sources are attributed in Table 2. By limiting WIBS data to only the FL3 data, the number concentration is expected to be significantly lower than total FBAP numbers often reported, but are used here for better correlation with the UV-APS due to similarities in fluorescence

excitation and emission bands (Foot et al., 2008; Pöhlker et al., 2012; Savage et al., 2017). For all LIF data, mean FBAP number concentrations were integrated from either 0.8 or 1.0 μm to 15 or 20 μm and are reported for each of seventeen geographic locations (see Table 2 and Fig. 6) for a comparison with our simulated mean number concentrations of fungal spores and bacteria produced from the model discussed here.. The FBAP observations show a peak in the number distribution at about 1 – 4 μm, irrespective of location or instrument. It has been previously suggested for several geographic

locations that the UV-APS and WIBS FL3 channel may yield lower limit proxies for fungal spores, due in part to the large number concentration of spores in this size range compared to other biological particles (Gosselin et al., 2016; Healy et al., 2014) .

**3. Results**

**3.1. Comparison of the EMAC fungal spore concentrations with spore count observations**

Our simulated, globally distributed, annual mean number concentrations using the three emission parameterizations are compared to the set of observations at various locations in Figure 1. Observational data are differentiated by biome as

defined by MODIS data, including the urban ecosystem. The modeled concentrations are overestimated by all three parameterizations, but least by HS, which agrees better with observed spore counts, with a correlation coefficient (R) of 0.22 and a median model-to-observation ratio of 6, compared to more than 100 for the other two parameterizations. Two additional metrics of model-observation agreement are presented in Figure 1: the modified normalized mean bias (MNMB), a measure of bias that is symmetric with respect to over-estimates and under-estimates (ranging between -2 to 2, and equal to




for a "perfect" model), and the fractional gross error (FGE), a measure of relative model error, ranging from 0 for a "perfect model" to a maximum value of 2, which behaves symmetrically with respect to under- and overestimation, without over-emphasizing outliers (Huijnen and Eskes, 2012).

From the three model versions, HS performs best on all three scores. SD and HU show similar MNMB and FGE, but SD shows a slightly negative correlation with the observations. Although we compare local measurements limited in their representativeness in time and space to the relatively coarse grid size (approximately 140 km) of the climatological model data, only the comparison with HS is satisfactory. Surprisingly, the SD derived fungal concentrations are the least comparable to observations despite the fact that the formulation of the emission parameterization is based partly on these

observations. This might be due to the change in the global ecosystems distribution, as we used the MODIS ecosystem, which the HS parameterization was based on, instead of the Olson distribution, which the SD parameterization was based on. The HU emission parameterization might be not suited for use in global modeling studies since it has been optimized for a regional modeling study over northern Europe. Differences in model physics, including the simulation distribution of precipitation, turbulent transport parameterizations, and parameterization of wet and dry removal, can also result in models

simulating different concentrations, given the same emissions, so these results cannot necessarily be extended to other atmospheric models.

Discrepancies between model and observations may be explained by an over-prediction of fungal spore sources via biases in the emission parameterization or long-range transport, or an under-prediction of the rate of removal by dry and wet

deposition. Additionally, as outlined by (Sesartic and Dallafior, 2011) and references therein, the observational data quality is limited and should be considered with caution. The methods used to measure actual spore concentrations may involve biases as well as problems related to the identification of fungal spores. As mentioned in section 2.3.1, (Sesartic and Dallafior, 2011) showed that many direct-counting spore techniques can significantly undercount spore number (i.e. by order of magnitude). Additionally, any culture-based methods have significant biases in that only a very small fraction of spore

species can be culturable in a given medium.

The total global emissions calculated here with HS (17 Tg yr$^{-1}$ corresponding to an average mass emission flux of 2.5 ng m$^{-2}$ s$^{-1}$) are within the range of uncertainties reported by (Després et al., 2012) and (Fröhlich-Nowoisky et al., 2016). Using the same HS parameterization, (Heald and Spracklen, 2009) and (Hoose et al., 2010) calculated higher totals, respectively 28 Tg

yr$^{-1}$ and 31 Tg yr$^{-1}$. This demonstrates the model sensitivity to the leaf area index dataset used for that purpose and the specific humidity calculated by the model. The total global emissions calculated using SD and HU are estimated to 86 Tg yr$^{-1}$ and 349 Tg yr$^{-1}$, respectively, which seem unrealistically high. Further, the comparison of fungal spore number fluxes calculated by (Sesartic and Dallafior, 2011) and by EMAC yields large discrepancies in magnitude and spatial distribution. This is most likely explained by large differences in the biome spatial distribution between MODIS and Olson data, leading



to the higher emissions calculated by EMAC when using the SD parameterization. Since the HS simulation shows a better fit to observations, we will show results only from this simulation hereafter.

Figure 3a shows that near-surface, annual mean number concentrations of fungal spores simulated by EMAC are typically
2.6 x $10^3$ $m^{-3}$, with a maximum over tropical forests of 6 x $10^3$ $m^{-3}$, matching the regions with largest emissions. These concentrations are one order of magnitude less than those calculated by (Heald and Spracklen, 2009) and (Spracklen and Heald, 2014) but within the range of concentrations reported by (Fröhlich-Nowoisky et al., 2009) and (Elbert et al., 2007). The spatial distribution shows large similarities with other modeling studies mentioned above. Our simulated zonal annual mean number concentrations of fungal spores, presented in Figure 3b, decrease with altitude from a maximum at the Equator
reaching up to 4 x $10^3$ to less than 100 $m^{-3}$ at 250 hPa, contributing, in theory, very little to global CCN concentrations (Pringle et al., 2010b). This is beyond the scope of this study, and will be the focus of a follow-up paper.

### 3.2. Correlation between observations and meteorological parameters

Meteorological variables affect the initial release of fungal spores into the atmosphere and the dispersal once airborne. Temperature and humidity affect the size of the source and control the release of some actively released fungal spores (Jones and Harrison, 2004). Their frequency and concentrations are equally dependent on geographical characteristics. Since the publications collected for this study do not always provide information on the meteorological parameters of the observational site, we investigate the effects of physical parameters such as temperature, specific humidity and leaf area
index as modeled by EMAC on the observed particle concentrations, taking into account the differentiation between ecosystems as defined by MODIS data. Interestingly, we found a strong difference between non-urban and urban observations through their correlations with the three parameters. Table 1 shows a correlation coefficient of 0.41 between the observations and specific humidity for the urban points and the same correlation between the non-urban observations and the leaf area index. This demonstrates the stronger effect of meteorological variables on urban sites, which could be taken into
account in the formulation of a more advanced fungal spore emission parameterization. Additional observational evidence is needed to support this hypothesis.

### 3.3. Bacteria concentrations

The total global emission source calculated by EMAC is 0.76 Tg $yr^{-1}$. This is within the range calculated by (Burrows et al., 2009a) (0.4 – 1.8 Tg $yr^{-1}$) with an older version of EMAC but using Olson vegetation types as ecosystem classification instead of the more recent, satellite-based MODIS vegetation distribution. Similar global emission estimates were reported by (Hoose et al., 2010) (0.7 Tg $yr^{-1}$). Although we calculate similar global annual mean number concentration (3.1 x $10^4$ $m^{-3}$) as in (Burrows et al., 2009a), we obtain differences in the geographical patterns and magnitudes of these concentrations in



Figure 4a, most notably over tropical forest regions where the concentrations are much lower than over other continental regions. The differences can only be due to the replacement of Olson by MODIS data and the resulting spatial distribution of emissions. Our spatial distribution and magnitudes of number concentrations are very similar to those presented by (Spracklen and Heald, 2014), who also used MODIS land cover classifications.

The zonal annual mean number concentrations of bacteria in Figure 4b are highest in the lower troposphere and decrease with altitude to reach about 100 m$^{-3}$ at 100 hPa. This might have a slight impact on global CCN concentrations as demonstrated by (Spracklen and Heald, 2014) as they found that bacteria contribute only 0.01% to global mean CCN concentrations near the surface, and that bacteria and fungal spores together contribute less than 1%.

A comparison of EMAC modeled PBAP concentrations (including bacteria, fungal spores and pollen) with FBAP observations from the IDEAS campaign in (Twohy et al., 2016) has shown that the model under-predicts observed concentrations in the free troposphere, while the most important PBAP category at high altitudes are bacteria, as indicated by Figure 4b.

**3.4. Pollen concentrations**

Figure. 5a and 5b show the zonal, annual mean number concentration distributions of pollen near the surface. The total global emission source calculated by EMAC is 44 Tg y$^{-1}$, and the average near-surface number concentration is 24 m$^{-3}$, being within the range of magnitudes reported by (Després et al., 2012) and references therein.

**3.5. Comparison of the EMAC simulated fungal spores and bacteria with FBAP observations**

Figure 6 shows a comparison between the observed fluorescent particle number and the modeled concentrations of bacteria and fungal spores. The comparison excludes two locations: Borneo and Nanjing, as the FBAP concentrations (0.2 cm$^{-3}$ and
2.09 cm$^{-3}$, respectively) are much larger than the total modeled bacterial and fungal spore concentrations (0.042 cm$^{-3}$ and 0.2 cm$^{-3}$). For the locations Saclay and Killaney, both UV-APS and WIBS observations are available, though, for the sake of clarity, only WIBS data are shown in Figure 6.

In general, the model results compare relatively well with the observations of total fluorescent particles in almost all
locations. The model concentrations are lower than FBAP concentrations in ca. 40% of the observations and are within a factor of two (high or low) in all cases. This model under-prediction is best observed in the summer season. The model predicts higher bacteria loadings than fungal spores (including in Nanjing) in almost all cases, with the only exceptions of Ucluelet (rural coastal) and the Amazon (near-pristine forest). Additionally, the simulated bacteria number concentrations in Figure 6 dominate the total PBAP concentrations for the locations where observations are available in winter, when very low





concentrations of fungal spores are expected. A possible explanation for this discrepancy is that bacteria emissions are assumed constant in time, representing inferred "background" emissions, while they may exhibit seasonal variations not taken into account in our model. Consequently, the differences in the seasonal bacteria concentrations for each location are related to transport and deposition patterns reproduced in the model. Fungal spore emissions, in contrast, are assumed to

include seasonal variability due to the seasonality of the leaf area index and the specific humidity, which are used as model inputs. Many reports support the observed seasonal cycle of fungal spore concentrations, which are typically highest in summer and early fall, but depends on latitude and ecosystem (Lacey, 1996; Lang-Yona et al., 2012; Manninen et al., 2014; Schumacher et al., 2013). Laboratory-based observations have shown that the WIBS FL3 channel utilized here is less efficient at detecting bacteria (Hernandez et al., 2016; Savage et al., 2017) and thus is likely to detect fungal spores or pollen

with relatively higher efficiency. The large size of pollen grains limits their ability to be detected by the WIBS, however, and so fungal spores are assumed to represent the largest fraction of biological particles detected by the FL3 WIBS channel used here. Moreover, evidence in certain campaigns suggests that these fluorescence channels correlate well with fungal spores (e.g., (Fernandez-Rodriguez et al., 2018; Healy et al., 2014; Huffman et al., 2012)). Other important technical considerations were discussed in Section 2.3.2.

The simulated fungal spores show the lowest concentrations during the winter season, and can be significant in the summer or dry seasons. This is especially the case for the Amazon and Ucluelet stations (and in Borneo, not shown) where fungal spores were shown by the model to dominate the total PBAP concentrations (Fig. 6). If we consider that the FBAPs observed in these locations mainly represent fungal spores, the model underestimates the fungal spores concentrations. This

contradicts results presented in Section 3.1, which shows a modeled overestimation with respect to spore counts measured by optical microscopy (Fig. 2). As mentioned, the comparison of modeled spore results with direct counts of spores may also be biased due to model inputs and because of the frequently observed undercounting of some collection and detection methods used for spore counting. FBAP and spore counts measured via optical microscopy were compared for two sets of new measurements shown here. The observed values from each method are reported here with the relative factor that the FBAP

concentration overcounts the spore counts from the optical technique shown in parentheses: Saclay - 0.048 x $10^6$ m$^{-3}$ (spore count), 0.088 x $10^6$ m$^{-3}$ (WIBS-4A; x1.8) and 0.027 x $10^6$ m$^{-3}$ (UV-APS; x0.6); Cyprus - 0.0015 x $10^6$ m$^{-3}$ (spore count), 0.0433 x $10^6$ m$^{-3}$ (WIBS-4A; x29). In several other cases, collocated measurements of FBAP and spore counts also show spore count to be lower than FBAP by a factor of ~2 (Fernandez-Rodriguez et al., 2018) to as ~10 (Healy et al., 2014; Huffman et al., 2012), again depending partially on instrumental parameters and differences in aerosols observed. Additional

direct comparison of FBAP concentrations with spore count concentrations summarized in this study (Table SI1 and Fig. 2) is not possible because of differences in locations and seasons. Therefore, the qualitative relationship between UV-LIF and other spore counting techniques has been demonstrated, but quantitative comparisons often show significant differences.





Significant uncertainties also still remain in the interpretation of the UV-LIF measurements. For example, UV-LIF measurements do not detect all spore types equally. (Hernandez et al., 2016; Savage et al., 2017) show differences in fluorescence profiles of a number of spore types and discuss that different instrument units detect particles with different detection efficiencies. This implies that both biological and instrumental factors can lead to differences in observed FBAP

concentrations. (Healy et al., 2014; Huffman et al., 2012) both discuss how certain types of spores may escape detection. In particular, (Fernandez-Rodriguez et al., 2018; Healy et al., 2014) discuss how the genus *Cladosporium* (among the most commonly observed spore type in many environments) correlated very poorly with fluorescent measurements, suggesting that dark-walled cell walls present in this type of spores may inhibit some types of real-time fluorescence detection. This spore type is a dominant spore type during dry weather, therefore it might be undercounted during the day and in certain

locations where such spore types are a high fraction of the spore number. In some cases the FL3 signal will also be influenced by non-biological particles (Savage et al., 2017), and so FBAP number concentrations from WIBS and UV-APS presented here should be used only for rough comparison.

### 3.6. Chemical aerosol mass composition simulated by EMAC

To evaluate the presence of PBAP particles (including bacteria, fungal spores and pollen) relative to other particle types, we perform a comparison to the dry aerosol composition simulated by EMAC, including dust, organic carbon, black carbon, sea salt and inorganic anthropogenic aerosols. The PBAP mass concentration is estimated from the number concentration assuming monodisperse and spherical particles (see Section 2) with a mass per particle of 0.52 pg, 33 pg and 250 ng for

bacteria, fungal spores and pollen, respectively. The model calculated mean total global aerosol mass concentration is approximately 65 $\mu$g m$^{-3}$. Global mean PBAP mass concentrations represent a total mass of approximately 5 $\mu$g m$^{-3}$, being dominated by pollen, which is less than 7% of the total simulated mass concentration. Desert dust is the main contributor to the total mass, as shown in figure 7A. When dust is excluded (Figure 7B) PBAPs account for 19% of the total mass concentration, while their contribution is 52% when both dust and sea salt are excluded (Figure 7C) Under the first

assumption and considering that dust is mostly concentrated over deserts, fungal spores and pollen mass concentrations contribute significantly to the total mass concentrations in central Africa (6% and 14%, respectively) and South America (5% and 11%, respectively). These values are much lower than the number and mass fractions of 80% reported by (Pöschl et al., 2010).

### 30   4. Discussion and Conclusions

We used the chemistry-climate model EMAC to simulate the total PBAPs present in the atmosphere including bacteria, fungal spores, and pollen. We compared the simulated fungal spore number concentrations, using three emission



parameterizations, to an updated data set of spore counts synthesized from the literature. Additional evaluation of the model simulated PBAP concentrations was performed using a comparison with a global data set of FBAP concentrations.

5 Bacteria were assumed to have an emission diameter of 4 μm, based on observations of bacteria-containing particles and clusters of bacteria. This allowed the inclusion of bacteria loadings in our model comparison with FBAP observations. Sensitivity runs with lower emission diameters (1 μm and 2 μm) showed similar near-surface global mean concentrations. The global distribution of bacteria concentrations is therefore not expected to change by varying the diameter.

10 From three fungal spore emission schemes, the HS parameterization showed the best fit to observations, although the model over-predicts the concentrations by up to a factor of 6 in some locations. Spore count observations were limited in time and space and are subject to several methodological issues. Therefore, these direct-count measurements cannot provide a rigorous evaluation of the model results. Bacteria and fungal spore concentrations predicted by the model compare well with FBAP observations from real-time measurement techniques. The overprediction/underprediction is estimated to be of a factor of 2 or less for all measurement locations. Modeled bacteria concentrations outnumber fungal spores in most 15 locations, particularly in winter, while simulated fungal spores might be underestimated in near-pristine environments. The FBAP concentrations used here (from UV-APS data and only FL3 for WIBS) are likely to underestimate fungal spores somewhat and dramatically underrate the abundance of bacteria in most locations. Most bacteria are not strongly fluorescent in the applied wavelength channels and are therefore underestimated. This might explain the differences when we expect high bacterial counts (e.g. (Savage et al., 2017)).

We conclude that fungal spore concentrations are underestimated by the model in most locations with respect to the FBAP data used, but overestimated with respect to direct count observations of spores. Furthermore, the bacteria emission scheme we applied does not represent seasonal nor diurnal variability, a weakness that needs to be addressed in future work. Further comparisons of our model results with long-term FBAP measurements, taking into account daily and seasonal variability, 25 might offer new opportunities to better constraint our model emission parameterizations. Improvements in the way FBAP data are analyzed will also allow for better separation of bacteria and fungal spores and will thus allow for better comparison with improved models in the future (Ruske et al., 2017).

We find that PBAPs contribute up to 19% to the total aerosol mass concentration globally after excluding dust aerosols. This 30 percentage is dominated primarily by pollen. Regionally, fungal spores may contribute significantly to the total aerosol mass. PBAP, fungal spores and pollen especially, account for a major part of the aerosol loading and are likely underestimated by our model. In the future we anticipate further adding an evaluation of future updates to the model with respect to modeled pollen concentrations in order to better characterize total PBAP mass.




**Acknowledgments**

We acknowledge the CEA/DAM/CBRN-E research programs and D. Baisnée (CEA) for providing the spore counts for Saclay and Cyprus. Unpublished WIBS measurements summarized in Table 2 were made possible through collaboration with many partners, including the following details. Measurements performed in Cyprus were partly supported by the EU-H2020 ACTRIS-2 project (European Union's Horizon 2020 research and innovation program, grant agreement No 654109) and the EU FP7-ENV-2013 BACCHUS project (grant Agreement 603445). Measurements in Ucluelet, Canada were led by Jixiao Li (Univ. Denver) and Ryan Mason (Univ. British Columbia) as a part of a NETCARE study (Mason et al., 2015) organized by Allan Bertram (Univ. British Columbia) and Jon Abbatt (Univ. Toronto) and with financial support from the NSERC CCAR program and from Environment Canada. Measurements in Barbados were organized and supported by the Max Planck Institute for Chemistry (Ulrich Pöschl), the University of Miami (Joseph Prospero), the Caribbean Institute for Meteorology and Hydrology (David Farrell), and were made possible with help from Christine Krentz (Univ. Denver), Mira Pöhlker, Bettina Weber, Maria Praß, Florian Ditas, Thomas Klimach, Petya Yordanova, and Christopher Pöhlker (all Max Planck Institute for Chemistry). ). UV-APS measurements in Paris (2010) in Paris were performed as a part of the MEGAPOLI campaign by Johannes Schneider (Max Planck Institute for Chemistry). Measurements in Saclay, France were performed as a part of the Biodetect 2014 study supported by CEA/DAM/CBRN-E, with assistance by Walfried Lassar (Univ. Denver). The authors gratefully acknowledge the stimulating discussions and exchange with Ulrich Pöschl and the Mainz Bioaerosol Laboratory. We thank Ulrich Pöschl, and Christopher Pöhlker for their critical reading of the manuscript. The University of Denver is acknowledged for financial support for Nicole Savage through the Phillipson Graduate Fellowship and for Kyle Pierce through a grant from the Undergraduate Research Center.



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



|  | Specific humidity | Temperature | Leaf Area Index |
|---|---|---|---|
| Urban Observations | **0.41** | 0.26 | -0.07 |
| Non-Urban Observations | 0.02 | 0.15 | **0.41** |

Table 1: Correlation table between observations and the modeled specific humidity, temperature and leaf area index. The values in bold are statistically significant (p < 0.05).

| Location | LON | LAT | ALT | Instrument | Start | End | Season | Site category | Publication | Size Range (μm) | Mean (10⁶m⁻³) |
|---|---|---|---|---|---|---|---|---|---|---|---|
| Amazon, Brazil | -60,2 | -2,58 | 110 | UV-APS | 03.02.2008 | 15.03.2008 | Winter | rainforest | Huffman et al., 2012) | >1 | 0,093 |
| Karlsruhe, Germany | 8,42 | 49,09 | 117 | WIBS-4A | 01.04.2010 | 01.04.2011 | Yearly | semi-rural | (Toprak and Schnaiter, 2013) | 0.8 -16 | 0,031 |
| Karlsruhe, Germany | 8,42 | 49,09 | 117 | WIBS-4A | 01.04.2010 | 01.07.2010 | Spring | semi-rural | (Toprak and Schnaiter, 2013) | 0.8 -16 | 0,029 |
| Karlsruhe, Germany | 8,42 | 49,09 | 117 | WIBS-4A | 01.07.2010 | 01.10.2010 | Summer | semi-rural | (Toprak and Schnaiter, 2013) | 0.8 -16 | 0,046 |
| Karlsruhe, Germany | 8,42 | 49,09 | 117 | WIBS-4A | 01.10.2010 | 01.01.2011 | Autumn | semi-rural | (Toprak and Schnaiter, 2013) | 0.8 -16 | 0,029 |
| Karlsruhe, Germany | 8,42 | 49,09 | 117 | WIBS-4A | 01.01.2011 | 01.04.2011 | Winter | semi-rural | (Toprak and Schnaiter, 2013) | 0.8 -16 | 0,019 |
| Colorado, USA | -105,15 | 39,16 | 2290 | UV-APS | 20.07.2011 | 22.08.2012 | Yearly | rural | (Schumacher et al., 2013) | >1 |  |
| Colorado, USA | -105,15 | 39,16 | 2290 | UV-APS | 01.03 | 31.05 | Spring | rural | (Schumacher et al., 2013) | >1 | 0,015 |
| Colorado, USA | -105,15 | 39,16 | 2290 | UV-APS | 01.06 | 31.08 | Summer | rural | (Schumacher et al., 2013) | >1 | 0,03 |
| Colorado, USA | -105,15 | 39,16 | 2290 | UV-APS | 01.09 | 30.11 | Autumn | rural | (Schumacher et al., 2013) | >1 | 0,017 |
| Colorado, USA | -105,15 | 39,16 | 2290 | UV-APS | 01.12 | 29.02 | Winter | rural | (Schumacher et al., 2013) | >1 | 0,0053 |
| Hyytiälä, Finland | 24,17 | 61,85 | 181 | UV-APS | 27.08.2009 | 17.04.2011 | Yearly | boreal forest | (Schumacher et al., 2013)) | >1 |  |
| Hyytiälä, Finland | 24,17 | 61,85 | 181 | UV-APS | 01.03 | 31.05 | Spring | boreal forest | (Schumacher et al., 2013) | >1 | 0,015 |
| Hyytiälä, Finland | 24,17 | 61,85 | 181 | UV-APS | 01.06 | 31.08 | Summer | boreal forest | (Schumacher et al., 2013) | >1 | 0,046 |
| Hyytiälä, Finland | 24,17 | 61,85 | 181 | UV-APS | 01.09 | 30.11 | Autumn | boreal forest | (Schumacher et al., 2013) | >1 | 0,027 |
| Hyytiälä, Finland | 24,17 | 61,85 | 181 | UV-APS | 01.12 | 29.02 | Winter | boreal forest | (Schumacher et al., 2013) | >1 | 0,004 |
| Killaney, Ireland | -9,5 | 52,05 | 34 | WIBS-4A | 02.08.2010 | 02.09.2010 | Summer | rural | (Healy et al., 2014) | >1 | 0,035 |
| Killaney, Ireland | -9,5 | 52,05 | 34 | UV-APS | 02.08.2010 | 02.09.2010 | Summer | rural | (Healy et al., 2014) | >1 | 0,015 |
| Mainz, Germany | 8,23 | 49,98 | 100 | UV-APS | 03.08.2006 | 04.12.2006 | Autumn | semi-urban | Huffman et al., 2010) | >1 | 0,027 |
| Borneo, Indonesia | 117,84 | 4,98 |  | WIBS-3 | 01.06.2008 | 31.07.2008 | Summer | forest | (Gabey et al., 2010) | 0.8-20 | 0,2 |
| Nanjing, China | 118,95 | 32,12 |  | WIBS-4A | 29.10.2013 | 15.11.2013 | Autumn | urban | Yu et al., 2013) | >1 | 2,09 |
| Puy de dôme, France | 2,96 | 45,43 | 1465 | WIBS-3 | 22.06.2010 | 03.07.2010 | Summer | mountain | (Gabey et al., 2013) | >1 | 0,095 |
| Manchester, UK | -2,25 | 53,48 |  | WIBS-3 | 04.12.2009 | 21.12.2009 | Winter | urban | (Gabey et al., 2011) | 0.8-20 | 0,11 |





| | | | | | | | | | | |
|---|---|---|---|---|---|---|---|---|---|---|
| Helsinki, Finland | 24,65 | 60,2 | | UV-APS | 02.02.2012 | 25.02.2012 | Winter | suburban | (Saari et al., 2015) | <1 | NA |
| Helsinki, Finland | 24,65 | 60,2 | | UV-APS | 16.06.2012 | 22.08.2012 | Summer | urban | (Saari et al., 2015) | >1 | 0,013 |
| Helsinki, Finland | 24,65 | 60,2 | | BIO-SCOUT | 02.02.2012 | 25.02.2012 | Winter | suburban | (Saari et al., 2015) | >1 | 0,01 |
| Helsinki, Finland | 24,65 | 60,2 | | BIO-SCOUT | 16.06.2012 | 22.08.2012 | Summer | urban | (Saari et al., 2015) | >1 | 0,028 |
| Munnar, India | 77,06 | 10,09 | 1605 | UV-APS | 01.06.2014 | 20.08.2014 | Summer | rural | (Valsan et al., 2016) | >1 | 0,017 |
| Ucluelet, Canada | -125,54 | 48,92 | 5 | WIBS-4A | 06.08.2013 | 28.08.2013 | Summer | rural/marine | unpublished | >1 | 0,059 |
| Paris, France | 2,35 | 48,85 | | UV-APS | 14.01.2010 | 15.02.2010 | Winter | urban | unpublished | >0.8 | 0,0276 |
| Saclay, France | 2,17 | 48,71 | | UV-APS | 16.06.2014 | 05.08.2014 | Summer | semi-urban | unpublished | >0.8 | 0,027 |
| Saclay, France | 2,17 | 48,71 | | WIBS-4A | 16.06.2014 | 05.08.2014 | Summer | semi-urban | unpublished | >0.8 | 0,088 |
| Cyprus | 33,06 | 35,03 | 550 | WIBS-4A | 01.04.2016 | 26.04.2016 | Spring | rural/mountain | unpublished | >0.8 | 0,0433 |
| Barbados | -59,43 | 13,16 | 5 | WIBS-4A | 16.07.2016 | 04.09.2016 | Summer | rural/marine | unpublished | >0.8 | 0,0951 |

Table 2: List of the FBAP observations.



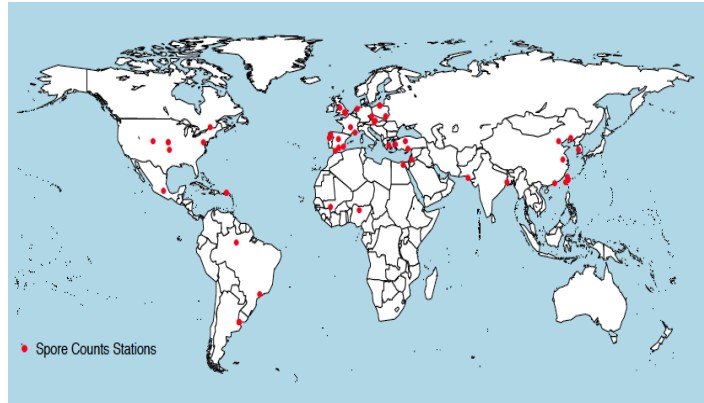

Fig. 1: Geographical locations of the fungal spore counts (List of locations is provided in Table SI1).

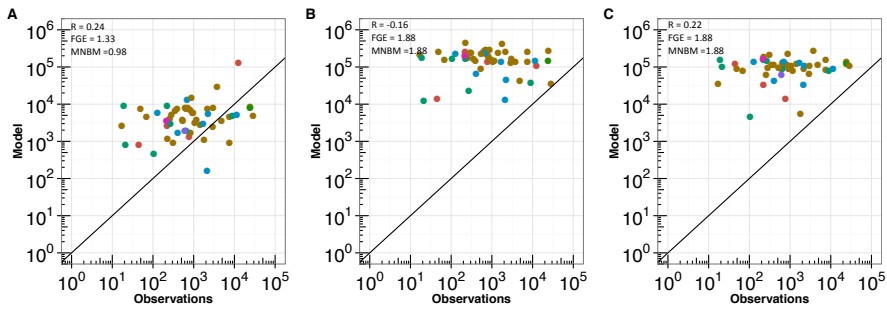

5    Fig. 2: Comparison between fungal spores number concentrations observed and simulated by EMAC using three emission parameterizations: A. (Heald and Spracklen, 2009), B. (Hummel et al., 2015) and C. (Sesartic and Dallafior, 2011). Point Colors depict the ecosystems of the observational stations as defined by MODIS. Units: m⁻³.




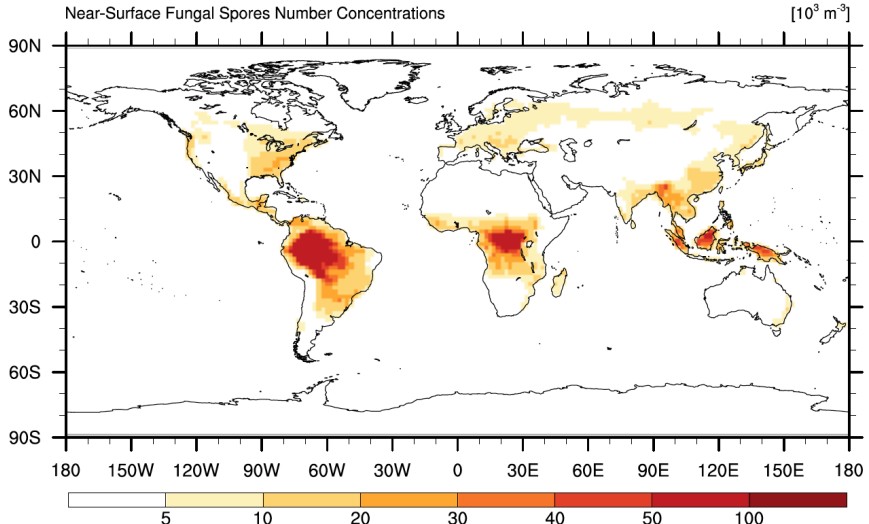

Fig. 3a: Modeled near-surface annual mean number concentration of fungal spores (in $10^3$ m$^{-3}$).



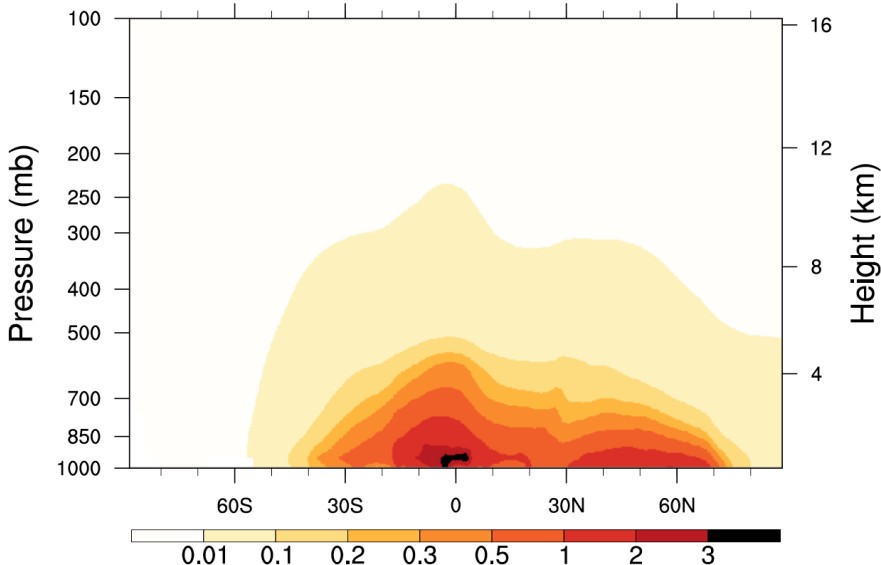

Fig. 3b: Zonal annual mean number concentrations of fungal spores (in $10^3$ m$^{-3}$).



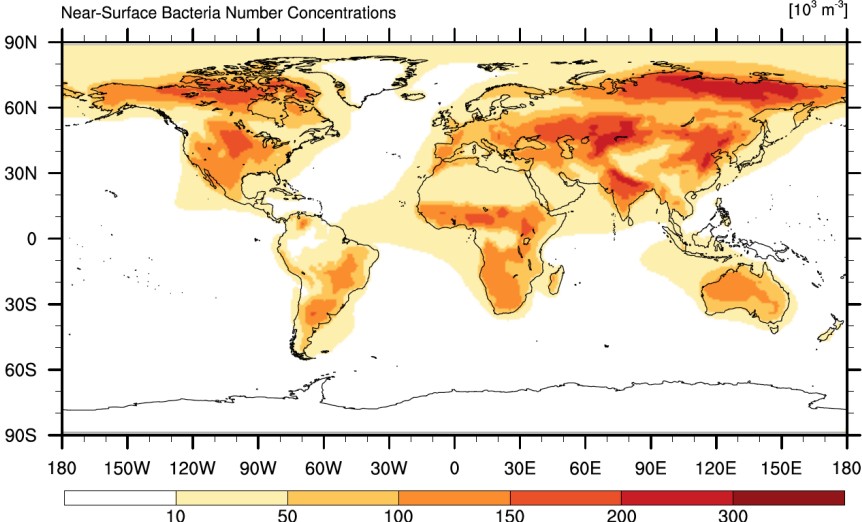

Fig. 4a: Modeled near-surface annual mean number concentration of bacteria (in $10^3$ m$^{-3}$).





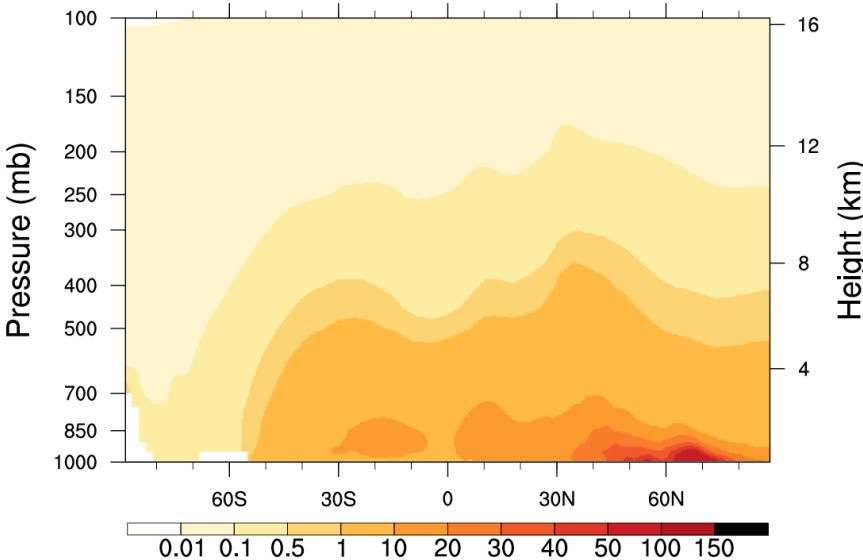

Fig. 4b: Zonal annual mean number concentrations of bacteria (in $10^3$ m$^{-3}$).





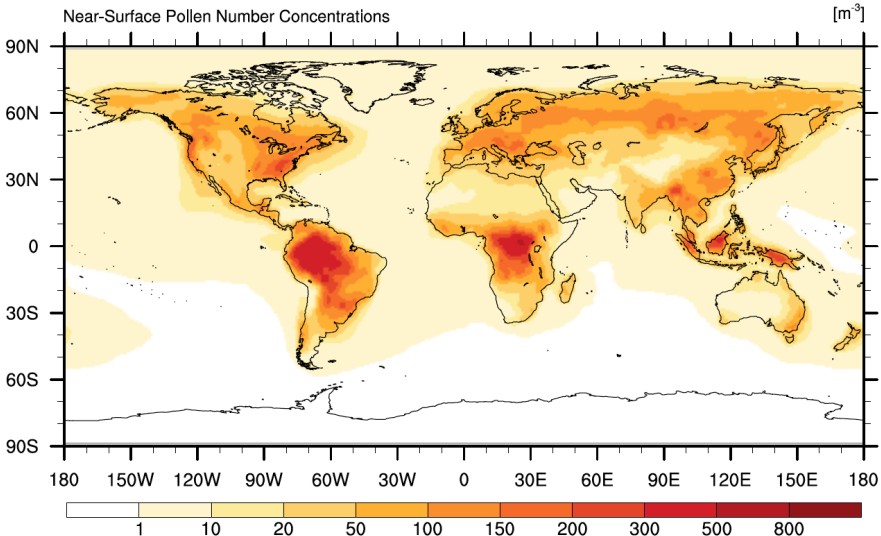

Fig. 5a: Modeled near-surface annual mean number concentration of pollen (in m$^{-3}$).





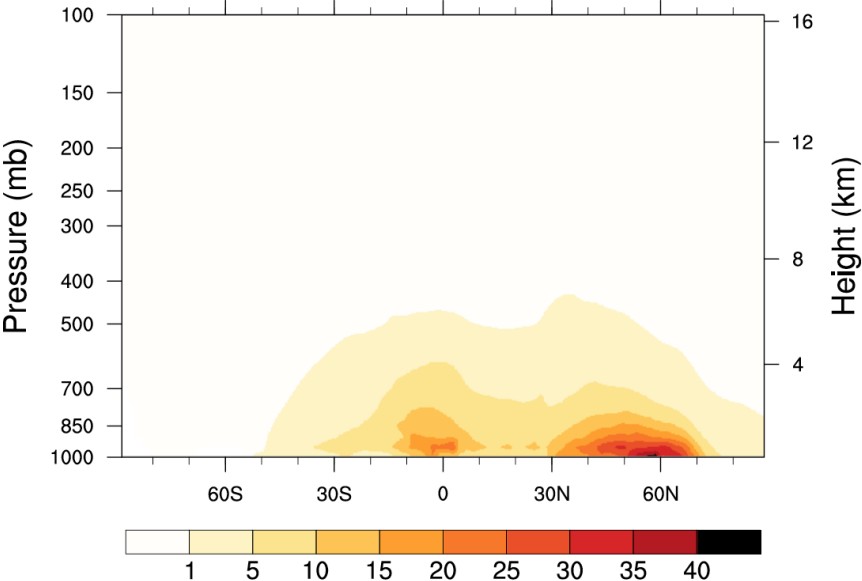

Fig. 5b: Zonal annual mean number concentrations of pollen (in m$^{-3}$).





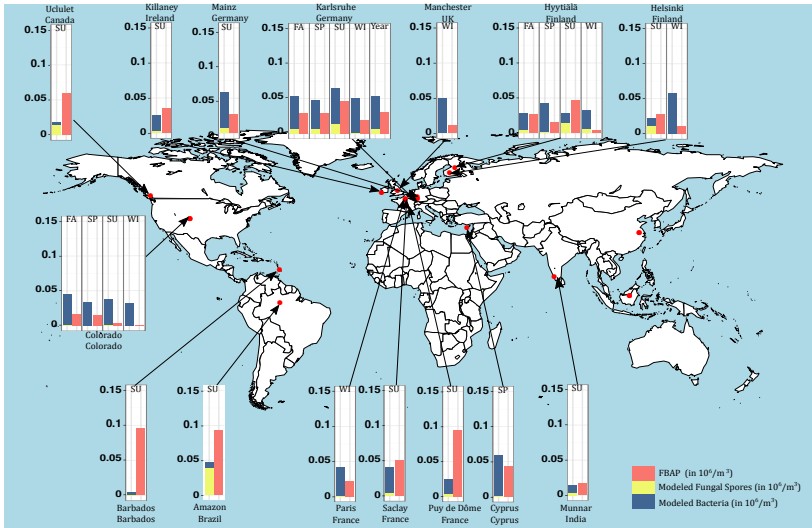

Fig. 6: Comparison between observed FBAP and modeled bacteria and fungal spores sampled for the campaigns described in Table SI 2. Units: $10^6 m^{-3}$.



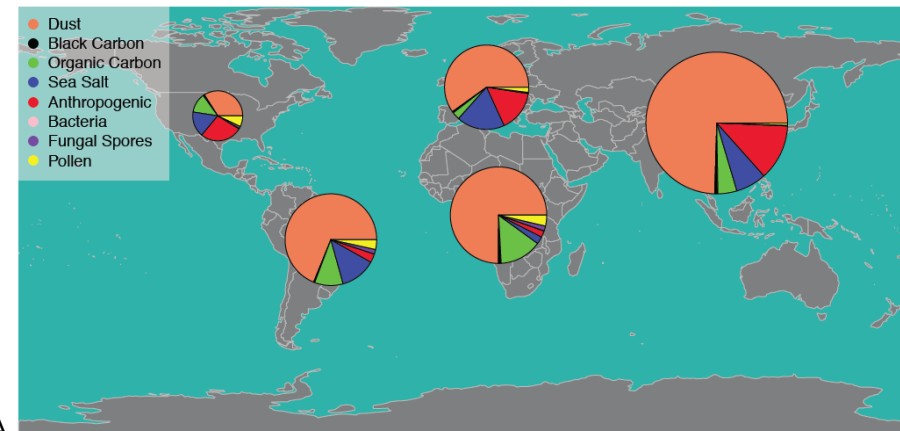

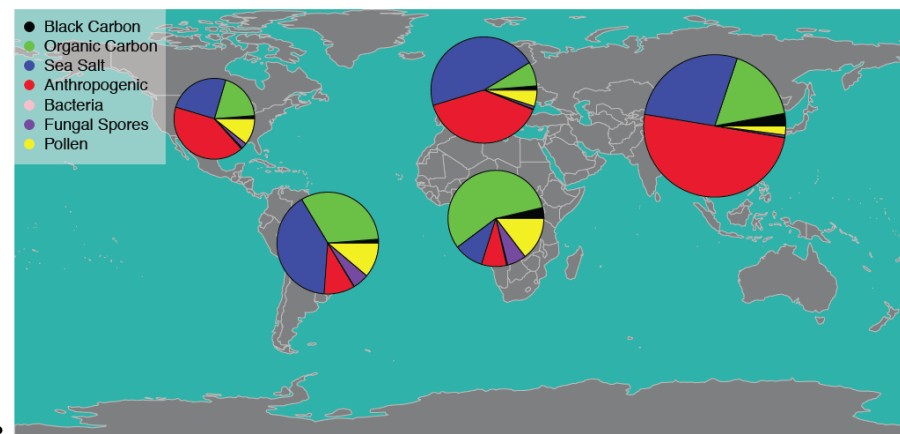





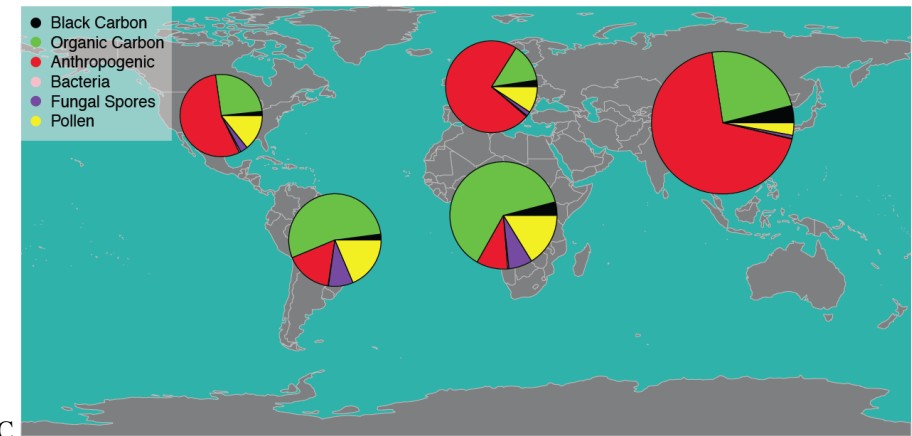

C

Figure 7: Near-surface annual mean chemical aerosol mass composition simulated by EMAC for five regions: North America (126°W – 72° W/30°N – 52° N), Europe (12°W – 36°E/34°N – 62°N), East Asia (100° - 144°E/20°N – 44°N), Central Africa (10°E – 40°E/10°S – 10°N) and South America (75°W – 35°W/30°S – 0°N). A. with dust, B. without dust, C. without dust and sea salt. The size of the pie chart is proportional to the total aerosol mass concentration.