# Peer review of "Global modeling of primary biological particle concentrations with the EMAC chemistry-climate model"

_Atmospheric Chemistry and Physics, 2018_

## Referee Comment (RC1) · Anonymous Referee #1 · 15 May 2018

This study explores the skill of three parameterizations for fungal spores, implemented in the EMAC model, to reproduce observed fungal spore counts, as well as (in combination with parameterized bacteria and pollen) fluorescence observations. Given the challenges in interpreting the observations (undercounting of spore counts, varying sensitivities of fluorescence), the study struggles to conclude as to the skill of these schemes as comparisons with the two datasets lead to opposite conclusions. Given this ambiguous result, I was disappointed that the authors did not pursue more exploration on the modeling side. I provide some specific suggestions below which would expand the utility of this study and ensure that it meets the standard for publication.

[Figure]

Interactive
comment

MAJOR

1. Expand modeling

- The results from none of the 3 fungal spore simulations is very satisfactory. Can the authors suggest (and possibly test?) improvements?

- Page 8,10: the authors claim that differences in bacteria from Burrows et al. may be the result of using the MODIS ecosystem distribution rather than the Olson distribution. This seems like something that could be easily confirmed with the model.

- Page 13: the authors highlight the deficiency of not including seasonal or diurnal variability for bacteria. Could they perform a simple sensitivity test to explore how imposing seasonal variation might impact their results?

2. Aerosol size assumptions

- Page 3, line 28: Why are fungal spores and bacteria treated as monodisperse? This seems an unrealistic assumption.

- Page 3, lines 32033: comment on the size dependence (if any) of these processes

- Page 5, line 17: 5 um seems large for fungal spores. Heald and Spracklen include fine and coarse mode particles, so this assumption does not seem consistent. Please comment.

- Page 8: While the authors claim that their results are not sensitive to the assumed size, it would certainly impact the conversion from number to mass. Might this help explain differences among previous fungal spore estimates discussed on page 9? If the authors feel these differences are not the result of assumed size, can they offer some explanation for these substantial differences? And why do the distribution and magnitude of bacteria agree better with previous studies than for fungal spores?

- Figures 3a, 4a, 5a seem to suggest more export of pollen and bacteria than spores (though this may be a false impression due to the color bar). Can the authors confirm

this by including global mean lifetime numbers for the 3 classes of PBAP? If lifetimes do differ, why is this the case when the authors indicated that removal processes are not dependent on size?

MINOR

1. Page 1, Line 19: measurements are spores, not all PBAP

2. Page 1, Line 23: meaning of "reflects a greater difference" is unclear. Compared to what? Observations? Or do the authors mean the ratio of bacteria to fungal spores varies more widely? Please modify text.

3. Page 1, lines 27-28: "of fungal spores and pollen", why not include bacteria in this sentence?

4. Page 6, line 9: what is the time span of the averaging? Is it possible to separate seasonal averages?

5. page 6, line 27: what is the upper size limit of these instruments?

6. Page 10, lines 24-26: why are the Borneo and Nanjing data exceptional? The authors need to justify why they would remove these from the comparisons

7. Section 3.5: the presentation of these results is a bit confusing. It would be helpful if the authors first discussed how many datasets are available for each season and commented on the observed seasonality before discussed the model performance.

8. Figure 6: It is very hard to see the data on this figure, suggest better use of scale (max value set to 0.1) to see data more clearly and using a color other than yellow/green which is hard to distinguish from white on the panels. The season labelling should be explained in the captions.

9. Page 11, line 1: unclear what "discrepancy" the authors are referring to. The measurements don't distinguish these two classes of PBAP.

10. Page 11, lines 25-28: clarify if these measurements are all for the same size ranges

11. Page 12, line 19: reference of justification for assumed mass per particle needed

12. Page 12, line 20: specify that these means are for the surface

13. Page 12, line 28: The Poschl et al. numbers are averages over the deployment and only for supermicron particles and so cannot be directly compared to annual means of all aerosols from the model. Suggest that you compare to relevant month, coarse fraction only. If the large difference holds when the correct time of year is compared can the authors speculate as to why there would be such a substantial difference?

———————————————

---

## Referee Comment (RC2) · Anonymous Referee #2 · 4 Jun 2018

Summary

This manuscript has developed a global emissions and transport model for primary biological aerosol particles (PBAP). There have been several other prior attempts to do this for all types of PBAP (e.g., Jacobson and Street, 2009) as well as individual types (e.g., fungal spores Heald and Spracklen 2009; bacteria Burrows et al 2009a/b). In this regard, the work is necessary but the manuscript itself does not make new advances in our simulations or understanding of global PBAP. In general, the manuscript has several major omissions of the data and methods driving the model, and this makes it not possible to interpret the results in any meaningful way. This paper requires major

revisions to be acceptable in ACP.

Major comments

1. The manuscript states to have simulated the three main types of PBAP, yet there is no detail in the manuscript about the pollen emissions. The authors need to decide if are going to retain the pollen section of the PBAP inventory. If the authors choose to continue to include pollen in their total PBAP assessment, there are several new sections that are absolutely necessary to understand what how the authors are simulating emissions and when and where they might be important. These include the following:

a. Add a section 2.2.3 for pollen. Section 2.2 is titled "PBAP emissions," with one subsection devoted to bacteria (2.2.1) and the other to fungal spores (2.2.2), yet there is no corresponding section on the pollen emissions. A section explaining the pollen emissions parameterization is absolutely required. Similarly, this should include the size distribution implemented in the model (as is for fungal spores and bacteria).

b. Provide some useful discussion on the pollen atmospheric distributions and their realism. For example, there is no discussion about the pollen emissions distributions that they simulate despite including a figure of pollen (Section 3.4). Specifically, their model simulates the highest concentrations of pollen in the tropics, which is inconsistent with the plant distribution of wind-driven pollen. Most plants in the tropics use insects or birds for transmission, so it is not expected that there would be high emissions in these locations. It is impossible to determine why this is without the explanation of the pollen emissions model (see point above).

c. There are ground-based pollen count observations of similar spatial sparsity to fungal spores. Why is a comparison of measured versus modeled not included?

d. In the final discussion, there is hardly any recommendations or future work regarding these emissions and improvements. Again, if the authors choose to keep pollen in there, a more rigorous discussion is required to explain the role of this specific type of

biological particle.

2. There is no information about the LAI distribution used in the model. This is rather important for the fungal spore discussion, as much of the explanation for the three different fungal spores is often tied back to differences in leaf area.

3. Section 3.2: This type of correlation seems rather obvious: modeled LAI is of course going to be lower in urban areas, so then another factor would have to compensate (and it would be a meteorological parameter). I'm not sure how this would not be taken into account already by the existing models. If the authors think that this is important, then they should explore this in greater detail.

4. Section 3.5: For the FBAP comparisons, why not also include the pollen and see if that improves any observations? The caveat of why FBAP may not (page 6, lines 20-21) or may (page 10, lines 10-13) is rather confusing.

Minor comments

1. The title should reflect the full acronym of PBAP (e.g., add the term "aerosol")

2. Abstract Lines 26-27: Needed?

3. Page 2 Line 19: fungal spores reference about being the most abundant and genetically diverse – this is a rather old reference, is there any more modern support for this idea?

4. Page 2 Line 35: Global and regional models are cited here, yet the papers primarily refer to global studies on fungal spores and bacteria. There is a wealth of literature out there on pollen, and this should be included if pollen is kept in the manuscript.

5. Page 5 line 31: Because the model was run without meteorological nudging, some brief reference to prior met evaluation of the model to indicate to readers that biases in the concentrations are not due to meteorological parameters such as temperature. 5. Page 4, section 2.2.1: More detail on the bacteria emissions, specifically the fact that

they are constant and are not simulate with any meteorological dependency should be made very clear in this section. This is discussed later in the manuscript (e.g., Section 3.5) but it would be clearer to provide more detail here such that a reader is not looking up all the references.

6. Page 4, line 29: what does "best estimate number fluxes" mean in this context?

7. Page 6, line 18: What are some examples of "highly fluorescent particles of non-biological origin," and would these be more likely to be observed in anthropogenically influenced areas?

8. Section 3.1: Note several references to Figure 1 that should be Figure 2, also, the fit metrics are not displayed as stated in the text (lines 12-13).

9. Section 3.1: Please clarify what model layers are used to compare to observed fungal spore counts, as this also may affect the model evaluation

---

## Short Comment (SC1) · 22 Jun 2018

1)In the parameterization of emission, recent field measurements of fluorescence show that there is an increase of primary biological particle during rainfall. It is likely that rain droplet splash would aerosolize more fungal spores and other biological particles. Your parameterization does not include a precipitation term to reflect that. Other authors (Huffman et al. 2013 doi:10.5194/acp-13-6151-2013; Schumacher et al. 2013, doi:10.5194/acp-13-11987-2013) have provided such a parameterization for rainfall. Could you comment on that ? 2)Burrows et al (2009) provided surface bacteria fluxes for terrestrial ecosystems. But over oceans, there is also bacterial fluxes. Is there an

account for ocean flux bacteria in your model ?

---

## Author Comment (AC1) · 15 Aug 2018

Response to Referee #1

**Comment**: This study explores the skill of three parameterizations for fungal spores, implemented in the EMAC model, to reproduce observed fungal spore counts, as well as (in combination with parameterized bacteria and pollen) fluorescence observations. Given the challenges in interpreting the observations (undercounting of spore counts, varying sensitivities of fluorescence), the study struggles to conclude as to the skill of these schemes as comparisons with the two datasets lead to opposite conclusions. Given this ambiguous result, I was disappointed that the authors did not pursue more exploration on the modeling side. I provide some specific suggestions below which would expand the utility of this study and ensure that it meets the standard for publication.

We thank the reviewer for the careful reading of the manuscript and helpful comments. We have revised the manuscript following the suggestions, as described below.

1. Expand modeling

**Comment**: The results from none of the 3 fungal spore simulations is very satisfactory. Can the authors suggest (and possibly test?) improvements?

**Response**: Based on the spore counts data we collected, we could not find any significant correlation between observations and the usual meteorological parameters influencing the spores release such as specific humidity, relative humidity, temperature, etc..(see Jones and Harrison, 2003); therefore we couldn't build any statistical relationship based on these data. More long-term observations reporting spore counts (in the form of time series) are needed to be able to build a new emission parametrization.
Nevertheless, since the parametrization proposed by Hoose et al. (2010) has been scaled to match the Heald and Spracklen (2009) emissions, we recommend applying an additional scaling factor of 6, which is the median of the ratio HS concentrations to the spore counts. This has been added to section 3.1.

- **Comment**: Page 8,10: the authors claim that differences in bacteria from Burrows et al. may be the result of using the MODIS ecosystem distribution rather than the Olson distribution. This seems like something that could be easily confirmed with the model.

**Response**: Indeed this can easily be confirmed with the model but not shown here since we know that Burrows et al. (2009b) used exactly the same model EMAC and setup (all processes included here), the only difference between setups is the ecosystem distribution.

**Comment**: Page 13: the authors highlight the deficiency of not including seasonal or diurnal variability for bacteria. Could they perform a simple

sensitivity test to explore how imposing seasonal variation might impact their results?

**Response**: The bacteria emission parameterization developed by Burrows et al. (2009b), used in this study, does not include any information about diurnal and seasonal variation. This was due to a lack of such observations for different ecosystems.

A new emission parameterization for bacteria will be the subject of a future independent work that will include the newly available observational data published since the publication of Burrows et al. (2009a).

2. Aerosol size assumptions
**Comment**: Page 3, line 28: Why are fungal spores and bacteria treated as monodisperse? This seems an unrealistic assumption.

**Response**: Fungal spores, bacteria and pollen are treated as monodisperse for the sake of consistency with previous modeling studies using the same assumption (Burrows et al. 2009b, Hoose et al. 2010a and 2010b, Haga et al. 2014, Hummel et al., 2015, Twohy et al. 2016, Hummel et al. 2018). Since the size distribution does not affect the removal rate in the model, there is no need to speculate about it, and add potentially misleading information, as the latter is not available from measurements.

**Comment**: Page 3, lines 32033: comment on the size dependence (if any) of these processes

**Response:** the size dependence of these processes, has been described by Tost et al. (2006) and Kerkweg et al. (2006), and the sensitivity of atmospheric transport to particle size for the EMAC model has been tested and described in detail in Burrows et al., 2013 and Kunkel et al. (2013).

**Comment**: Page 5, line 17: 5 um seems large for fungal spores. Heald and Spracklen include fine and coarse mode particles, so this assumption does not seem consistent. Please comment.

**Response**: As mentioned earlier, we used the emission parametrization proposed by Hoose et al. (2010) adapted from the emission estimates of Heald and Spracklen (2009). This emission parametrization has been calculated based on the assumption of a mean spore diameter of 5um.

**Comment:** Page 8: While the authors claim that their results are not sensitive to the assumed size, it would certainly impact the conversion from number to mass. Might this help explain differences among previous fungal spore estimates discussed on page 9? If the authors feel these differences are not the result of assumed size, can they offer some explanation for these substantial differences? And why do the distribution and magnitude of bacteria agree better with previous studies than for fungal spores?

**Response**: We are not certain if the referee is referring to the comparisons with previous estimates of global total fungal spore emissions, at the bottom of page 8, or the comparisons of simulated fungal spore number concentrations with observed fungal spore number concentrations (spore counts), at the top of page 9.

On page 8 (l. 26-33), we compare with previous emission estimates that were reported on a mass basis in earlier studies. Compared with 17 Tg yr-1 calculated in this study, Heald and Spracklen (2009) calculated 28 Tg yr-1, and Hoose et al. (2010) calculated 31 Tg yr-1, all when using the same emission parameterization. Hoose et al. (2010) used a mean spore diameter of 5um and Heald and Spracklen (2009) used the same diameter in the coarse mode.

These differences with previous fungal spore estimates are explained by the physical parameters in the emission scheme, i.e., Leaf area index and simulated surface humidity in the different host models.

By contrast, the different emission schemes formulate emissions differently, producing large discrepancies in simulated emissions between the different schemes.

On page 9, we discuss the comparison with number concentrations simulated when using these different emissions schemes, versus observed spore counts (number concentrations), shown in Figure 2. This comparison does not depend on the conversion of simulated number to simulated mass.

**Comment**: Figures 3a, 4a, 5a seem to suggest more export of pollen and bacteria than spores (though this may be a false impression due to the color bar). Can the authors confirm this by including global mean lifetime numbers for the 3 classes of PBAP? If lifetimes do differ, why is this the case when the authors indicated that removal processes are not dependent on size?

**Response**: We could say that, actually, the color bars are not easily comparable. The scales of fungi and bacteria must be multiplied by a factor of 1000, i.e. different from the scale of pollen. Thus, pollen is actually the aerosol category that is less transported. Based on the comments of Referee#2, we preferred to move the pollen figures to the SI.

Nevertheless, the reviewer's comments have helped us to realize that the dependence of transport and removal on particle size was not sufficiently explained in the original manuscript.

The export of particles simulated by the model varies as a function of both particle size and the geographic location of the emissions, which determines the atmospheric transport and removal processes the particles experience.

However, the differences in atmospheric residence time and export that are associated with particle size are on the order of a factor of ca. 1.5 – 2.5 when varying the particle size between 1 um and 10 um (Burrows et al., 2013, Figure

1, reproduced below for convenience).  This is much smaller than the model-observation differences shown in Figure 2, which are frequently 2-3 orders of magnitude. A clarification has been added to the revised manuscript.

[Figure]

MINOR

**Comment**: 1. Page 1, Line 19: measurements are spores, not all PBAP

**Response**: This has been corrected.

**Comment**: 2. Page 1, Line 23: meaning of "reflects a greater difference" is unclear. Compared to what? Observations? Or do the authors mean the ratio of bacteria to fungal spores varies more widely? Please modify text.

**Response:** This part of the sentence has been removed for the sake of clarity.

**Comment**: 3. Page 1, lines 27-28: "of fungal spores and pollen", why not include bacteria in this sentence?

**Response**: The contribution of the global bacteria mass concentrations to the total aerosol mass is too low (less than 1%) to be cited here.

**Comment**: 4. Page 6, line 9: what is the time span of the averaging? Is it possible to separate seasonal averages?

**Response**: Here the "averaging" means the averages over 4 years simulation. We provide a climatological value for each period of observation.  All model data are sampled according to the time period of each observation.

**Comment: 5. page 6, line 27: what is the upper size limit of these instruments?**

The upper size limit of the UV-APS and WIBS is nominally 20 um, though in practice the inlet design of an individual measurement site frequently lowers the upper size point somewhat.

**Comment**: 6. Page 10, lines 24-26: why are the Borneo and Nanjing data exceptional? The authors need to justify why they would remove these from the comparisons

**Response**: The Nanjing data reported too high concentrations that could be attributed partially to domestic pollution rather than biological particles (see the discussion related to these measurements). This has been added to the revised manuscript. For Borneo, we added it to the figure 6 with a different scale, and it has been included to the discussion.

**Comment**: 7. Section 3.5: the presentation of these results is a bit confusing. It would be helpful if the authors first discussed how many datasets are available for each season and commented on the observed seasonality before discussed the model performance.

**Response**: our point in this section is not to compare the seasonality of the model to observations. We believe that seasonality is not relevant here. The observations are only available for these specific time periods and our main objective is to compare the model concentrations to the available observations.

**Comment**: 8. Figure 6: It is very hard to see the data on this figure, suggest better use of scale (max value set to 0.1) to see data more clearly and using a color other than yellow/green which is hard to distinguish from white on the panels. The season labelling should be explained in the captions.

**Response**: We corrected the scale and changed the colors of the figure as the referee suggested.

**Comment**: 9. Page 11, line 1: unclear what "discrepancy" the authors are referring to. The measurements don't distinguish these two classes of PBAP.

**Response:** indeed the measurements don't distinguish between the two classes, but we expect from the measurements to see more fungal spores than bacteria (given all the uncertainties related to these measurements)

**Comment:** 10. Page 11, lines 25-28: clarify if these measurements are all for the same size ranges .

**Response**: Indeed, these measurements, as explained earlier in the paragraph and in Table 2, are for the same ranges.

**Comment:** 11. Page 12, line 19: reference of justification for assumed mass per particle needed

**Response:** The references have been added in the revised manuscript

**Comment**: 12. Page 12, line 20: specify that these means are for the surface

**Response**: The word "surface" has been added to the text.

**Comment**: 13. Page 12, line 28: The Poschl et al. numbers are averages over the deployment and only for supermicron particles and so cannot be directly compared to annual means of all aerosols from the model. Suggest that you compare to relevant month, coarse fraction only. If the large difference holds when the correct time of year is compared can the authors speculate as to why there would be such a substantial difference?

**Response**: We agree with the referee that this comparison is not appropriate therefore it has been removed.

---

## Author Comment (AC2) · 15 Aug 2018

Response to Referee#2

We thank the reviewer for the careful reading of the manuscript and helpful comments. We have revised the manuscript following the suggestions, as described below.

Summary
This manuscript has developed a global emissions and transport model for primary biological aerosol particles (PBAP). There have been several other prior attempts to do this for all types of PBAP (e.g., Jacobson and Street, 2009) as well as individual types (e.g., fungal spores Heald and Spracklen 2009; bacteria Burrows et al 2009a/b). In this regard, the work is necessary but the manuscript itself does not make new advances in our simulations or understanding of global PBAP. In general, the manuscript has several major omissions of the data and methods driving the model, and this makes it not possible to interpret the results in any meaningful way. This paper requires major revisions to be acceptable in ACP.

Major comments

1. The manuscript states to have simulated the three main types of PBAP, yet there is no detail in the manuscript about the pollen emissions. The authors need to decide if are going to retain the pollen section of the PBAP inventory. If the authors choose to continue to include pollen in their total PBAP assessment, there are several new sec- tions that are absolutely necessary to understand what how the authors are simulating emissions and when and where they might be important. These include the following:

**Comment**: a. Add a section 2.2.3 for pollen. Section 2.2 is titled "PBAP emissions," with one subsection devoted to bacteria (2.2.1) and the other to fungal spores (2.2.2), yet there is no corresponding section on the pollen emissions. A section explaining the pollen emissions parameterization is absolutely required. Similarly, this should include the size distribution implemented in the model (as is for fungal spores and bacteria).

**Response**: We apologize for this omission in the text. The section has been added to the revised manuscript.

**Comment:** b. Provide some useful discussion on the pollen atmospheric distributions and their re- alism. For example, there is no discussion about the pollen emissions distributions that they simulate despite including a figure of pollen (Section 3.4). Specifically, their model simulates the highest concentrations of pollen in the tropics, which is inconsistent with the plant distribution of wind-driven pollen. Most plants in the tropics use insects or birds for transmission, so it is not expected that there would be high emissions in these locations. It is impossible to determine why this is without the explanation of the pollen emissions model (see point above).

**Response**: The emissions depend on LAI similar to fungal spores, therefore, it should not be unexpected that concentrations are high in the tropics. It is difficult to discuss the realism of the global pollen distribution, as an important effort is needed to collect a global database for pollen. The Jacobson and Streets (2009) provided the first pollen parametrization to be used for a global model. This parameterization has not been evaluated against observations for any global model, and has been mostly used by the community to provide a global distribution for pollen, despite of its many deficiencies (e.g absence of the plant phenology and the land cover type). This was not the objective of the present study, as we focused on fungal spore distribution. In the discussion section, we recommend the use of a more recent parametrization developed and evaluated by Wozniak and Steiner (2017). For these reasons, we decided to remove Figures 5a and 5b from the main text and added them to the Supplementary Information, but we keep section 3.4, to be able to discuss the contribution of pollen to the total mass aerosol composition.

**Comment**: c. There are ground-based pollen count observations of similar spatial sparsity to fungal spores. Why is a comparison of measured versus modeled not included?

**Response**: Our purpose in this study is to focus on fungal spores parametrizations and to less extent bacteria. Pollen has been added to model as an additional PBAP only in order to estimate their contribution to the total aerosol mass (see section 3.6). A further modeling study using the new pollen emission parametrization proposed by Wozniak and Steiner (2017) will be the object of a following study including comparison with the available pollen count observations, especially in Europe, where a large pollen counts database is available.

**Comment:** d. In the final discussion, there is hardly any recommendations or future work regarding these emissions and improvements. Again, if the authors choose to keep pollen in there, a more rigorous discussion is required to explain the role of this specific type of biological particle.

**Response**: Recommendations have been added to the discussion section.

**Comment:** 2. There is no information about the LAI distribution used in the model. This is rather important for the fungal spore discussion, as much of the explanation for the three different fungal spores is often tied back to differences in leaf area.

**Response**: Both the HS and HU parametrization in our simulations use the same LAI distribution (as well as the pollen parametrization). Information about the LAI distribution used in our simulation has been added to the revised manuscript in section 2.1.

**Comment:** 3. Section 3.2: This type of correlation seems rather obvious: modeled LAI is of course going to be lower in urban areas, so then another factor would have to compensate (and it would be a meteorological parameter). I'm not

sure how this would not be taken into account already by the existing models. If the authors think that this is important, then they should explore this in greater detail.

**Response**: We found these results interesting because they show the importance of the difference between urban and non-urban observations and their relationship with these physical parameters (usually not measured). This information could be taken into account potentially in future observations. Unfortunately, we could not find any observation publication reporting about these differences.

**Comment**: 4. Section 3.5: For the FBAP comparisons, why not also include the pollen and see if that improves any observations? The caveat of why FBAP may not (page 6, lines 20-21) or may (page 10, lines 10-13) is rather confusing.

**Response**: As shown in figure 5a, the pollen number concentrations are much lower that bacteria and fungal spores concentrations, therefore their contribution to the total PBAP concentrations will be very low. Besides, as explained in Page 10, the upper size limit of the UV-APS and WIBS is nominally 20 um, though in practice the inlet design of an individual measurement site frequently lowers the upper size point somewhat, therefore pollen can not be included for comparison with FBAP observations.

Minor comments

**Comment**: 1. The title should reflect the full acronym of PBAP (e.g., add the term "aerosol")

**Response**: The word has been added to the title.

**Comment**: 2. Abstract Lines 26-27: Needed?

**Response**: We believe that these lines summarize one of the main findings of this paper, therefore they should be kept in the abstract.

**Comment**: 3. Page 2 Line 19: fungal spores reference about being the most abundant and genetically diverse – this is a rather old reference, is there any more modern support for this idea?

Response: The reference to the work of Fröhlich-Nowoisky et al. (2009) has been added

**Comment:** 4. Page 2 Line 35: Global and regional models are cited here, yet the papers primarily refer to global studies on fungal spores and bacteria. There is a wealth of literature out there on pollen, and this should be included if pollen is kept in the manuscript.

**Response**: Actually, Jacobson and Streets (2009), whose pollen parametrization has been used here and Hoose et al. (2010) include both pollen modeling.

**Comment**: 5. Page 5 line 31: Because the model was run without meteorological nudging, some brief reference to prior met evaluation of the model to indicate to readers that biases in the concentrations are not due to meteorological parameters such as temperature.

**Response**: References to the model evaluation has been added in section 2.1., as recommended by the referee.

**Comment:** 5. Page 4, section 2.2.1: More detail on the bacteria emissions, specifically the fact that they are constant and are not simulate with any meteorological dependency should be made very clear in this section. This is discussed later in the manuscript (e.g., Section 3.5) but it would be clearer to provide more detail here such that a reader is not looking up all the references.

**Response:** For the sake of clarity, we provide a table of the best-estimates fluxes in the Supplementary Material, and a reference to it in section 2.2.1.

**Comment:** 6. Page 4, line 29: what does "best estimate number fluxes" mean in this context?

**Response:** "best-estimate" is used considering the optimization method used by Burrows et al. (2009b) for the emission estimates to fit the observed number concentrations.

**Comment**: 7. Page 6, line 18: What are some examples of "highly fluorescent particles of non- biological origin," and would these be more likely to be observed in anthropogenically influenced areas?

**Response**: This sentence "such as certain kinds of aged brown SOA, diesel soot particles and some HULIS types" has been added to the text.

**Comment**: 8. Section 3.1: Note several references to Figure 1 that should be Figure 2, also, the fit metrics are not displayed as stated in the text (lines 12-13).

**Response**: This has been corrected in the revised manuscript.

**Comment**: 9. Section 3.1: Please clarify what model layers are used to compare to observed fungal spore counts, as this also may affect the model evaluation.

**Response**: The word "surface" has been added to the mean number concentrations.

---

## Author Comment (AC3) · 15 Aug 2018

**Response to A. Robichaud**

**Comment**: In the parameterization of emission, recent field measurements of fluorescence show that there is an increase of primary biological particle during rainfall. It is likely that rain droplet splash would aerosolize more fungal spores and other biological particles. Your parameterization does not include a precipitation term to reflect that. Other authors (Huffman et al. 2013 doi:10.5194/acp-13-6151-2013; Schumacher et al. 2013, doi:10.5194/acp-13-11987-2013) have provided such a parameterization for rainfall. Could you comment on that?

**Response**: Indeed, our three emission parameterizations do not include a precipitation term. Our purpose is to test the available emission parameterizations using our model. Based on the assessment of the uncertainties related to this type of measurements, described in detail in the manuscript, we cannot build at this stage a new parameterization based on these findings. The rainfall-induced emissions have also been shown to be variable as a function of geography and season.  I.e. Schumacher et al. (2013) shows relatively strong post-rainfall emissions of what the authors suggest are fungal spores, but primarily during the summer months in relatively arid Colorado forest.  The same paper shows only a minimal effect in the boreal forest of Finland. In even stronger contrast, no rain-related enhancement of fluorescent (biological) particles was observed at the Amazon site and also many others (Huffman et al. 2012).  Given the complexity of the ability to predict rainfall-related emissions, we did not explore rainfall with this model.

**Comment**: 2) Burrows et al (2009) provided surface bacteria fluxes for terrestrial ecosystems. But over oceans, there is also bacterial fluxes. Is there an account for ocean flux bacteria in your model ?

**Response**: In table SI1 we provide the emission fluxes derived from Burrows et al (2009b). Indeed the sea emission flux is set to zero.  As explained in Burrows et al. (2009a), there is no observational evidence, at least at the time of publication, about total bacteria concentration in remote marine air. The references suggest rather that dust plumes are the major source of bacteria in marine air (see section 9 "bacteria in the marine air").

---

## Author Comment (AC4) · 15 Aug 2018

**Response to Referee #1**

**Comment**: This study explores the skill of three parameterizations for fungal spores, implemented in the EMAC model, to reproduce observed fungal spore counts, as well as (in combination with parameterized bacteria and pollen) fluorescence observations. Given the challenges in interpreting the observations (undercounting of spore counts, varying sensitivities of fluorescence), the study struggles to conclude as to the skill of these schemes as comparisons with the two datasets lead to opposite conclusions. Given this ambiguous result, I was disappointed that the authors did not pursue more exploration on the modeling side. I provide some specific suggestions below which would expand the utility of this study and ensure that it meets the standard for publication.

We thank the reviewer for the careful reading of the manuscript and helpful comments. We have revised the manuscript following the suggestions, as described below.

1. Expand modeling

**Comment**: The results from none of the 3 fungal spore simulations is very satisfactory. Can the authors suggest (and possibly test?) improvements?

**Response**: Based on the spore counts data we collected, we could not find any significant correlation between observations and the usual meteorological parameters influencing the spores release such as specific humidity, relative humidity, temperature, etc..(see Jones and Harrison, 2003); therefore we couldn't build any statistical relationship based on these data. More long-term observations reporting spore counts (in the form of time series) are needed to be able to build a new emission parametrization.

Nevertheless, since the parametrization proposed by Hoose et al. (2010) has been scaled to match the Heald and Spracklen (2009) emissions, we recommend applying an additional scaling factor of 6, which is the median of the ratio HS concentrations to the spore counts. This has been added to section 3.1.

**Comment**: Page 8,10: the authors claim that differences in bacteria from Burrows et al. may be the result of using the MODIS ecosystem distribution rather than the Olson distribution. This seems like something that could be easily confirmed with the model.

**Response**: Indeed this can easily be confirmed with the model but not shown here since we know that Burrows et al. (2009b) used exactly the same model EMAC and setup (all processes included here), the only difference between setups is the ecosystem distribution.

**Comment**: Page 13: the authors highlight the deficiency of not including seasonal or diurnal variability for bacteria. Could they perform a simple sensitivity test to explore how imposing seasonal variation might impact their results?

5 **Response**: The bacteria emission parameterization developed by Burrows et al. (2009b), used in this study, does not include any information about diurnal and seasonal variation. This was due to a lack of such observations for different ecosystems. A new emission parameterization for bacteria will be the subject of a future independent work that will include the newly available observational data published since the publication of Burrows et al. (2009a).

10 2. Aerosol size assumptions

**Comment**: Page 3, line 28: Why are fungal spores and bacteria treated as monodisperse? This seems an unrealistic assumption.

**Response**: Fungal spores, bacteria and pollen are treated as monodisperse for the sake of consistency with previous
15 modeling studies using the same assumption (Burrows et al. 2009b, Hoose et al. 2010a and 2010b, Haga et al. 2014, Hummel et al., 2015, Twohy et al. 2016, Hummel et al. 2018). Since the size distribution does not affect the removal rate in the model, there is no need to speculate about it, and add potentially misleading information, as the latter is not available from measurements.

**Comment**: Page 3, lines 32033: comment on the size dependence (if any) of these processes

**Response:** the size dependence of these processes, has been described by Tost et al. (2006) and Kerkweg et al. (2006), and the sensitivity of atmospheric transport to particle size for the EMAC model has been tested and described in detail in
25 Burrows et al., 2013 and Kunkel et al. (2013).

**Comment**: Page 5, line 17: 5 um seems large for fungal spores. Heald and Spracklen include fine and coarse mode particles, so this assumption does not seem consistent. Please comment.

30 **Response**: As mentioned earlier, we used the emission parametrization proposed by Hoose et al. (2010) adapted from the emission estimates of Heald and Spracklen (2009). This emission parametrization has been calculated based on the assumption of a mean spore diameter of 5um.

**Comment:** Page 8: While the authors claim that their results are not sensitive to the assumed size, it would certainly impact the conversion from number to mass. Might this help explain differences among previous fungal spore estimates discussed on page 9? If the authors feel these differences are not the result of assumed size, can they offer some explanation for these substantial differences? And why do the distribution and magnitude of bacteria agree better with previous studies than for fungal spores?

**Response**: We are not certain if the referee is referring to the comparisons with previous estimates of global total fungal spore emissions, at the bottom of page 8, or the comparisons of simulated fungal spore number concentrations with observed fungal spore number concentrations (spore counts), at the top of page 9.

On page 8 (l. 26-33), we compare with previous emission estimates that were reported on a mass basis in earlier studies. Compared with 17 Tg yr-1 calculated in this study, Heald and Spracklen (2009) calculated 28 Tg yr-1, and Hoose et al. (2010) calculated 31 Tg yr-1, all when using the same emission parameterization. Hoose et al. (2010) used a mean spore diameter of 5um and Heald and Spracklen (2009) used the same diameter in the coarse mode.

These differences with previous fungal spore estimates are explained by the physical parameters in the emission scheme, i.e., Leaf area index and simulated surface humidity in the different host models.

By contrast, the different emission schemes formulate emissions differently, producing large discrepancies in simulated emissions between the different schemes.

On page 9, we discuss the comparison with number concentrations simulated when using these different emissions schemes, versus observed spore counts (number concentrations), shown in Figure 2. This comparison does not depend on the conversion of simulated number to simulated mass.

**Comment**: Figures 3a, 4a, 5a seem to suggest more export of pollen and bacteria than spores (though this may be a false impression due to the color bar). Can the authors confirm this by including global mean lifetime numbers for the 3 classes of PBAP? If lifetimes do differ, why is this the case when the authors indicated that removal processes are not dependent on size?

**Response**: We could say that, actually, the color bars are not easily comparable. The scales of fungi and bacteria must be multiplied by a factor of 1000, i.e. different from the scale of pollen. Thus, pollen is actually the aerosol category that is less transported. Based on the comments of Referee#2, we preferred to move the pollen figures to the SI.

Nevertheless, the reviewer's comments have helped us to realize that the dependence of transport and removal on particle size was not sufficiently explained in the original manuscript.

The export of particles simulated by the model varies as a function of both particle size and the geographic location of the emissions, which determines the atmospheric transport and removal processes the particles experience.

However, the differences in atmospheric residence time and export that are associated with particle size are on the order of a factor of ca. 1.5 – 2.5 when varying the particle size between 1 um and 10 um (Burrows et al., 2013, Figure 1, reproduced below for convenience). This is much smaller than the model-observation differences shown in Figure 2, which are frequently 2-3 orders of magnitude. A clarification has been added to the revised manuscript.

[Figure]

MINOR

**Comment**: 1. Page 1, Line 19: measurements are spores, not all PBAP

**Response**: This has been corrected.

**Comment**: 2. Page 1, Line 23: meaning of "reflects a greater difference" is unclear. Compared to what? Observations? Or do the authors mean the ratio of bacteria to fungal spores varies more widely? Please modify text.

**Response:** This part of the sentence has been removed for the sake of clarity.

**Comment**: 3. Page 1, lines 27-28: "of fungal spores and pollen", why not include bacteria in this sentence?

**Response**: The contribution of the global bacteria mass concentrations to the total aerosol mass is too low (less than 1%) to be cited here.

**Comment**: 4. Page 6, line 9: what is the time span of the averaging? Is it possible to separate seasonal averages?

**Response**: Here the "averaging" means the averages over 4 years simulation. We provide a climatological value for each period of observation. All model data are sampled according to the time period of each observation.

**Comment: 5. page 6, line 27: what is the upper size limit of these instruments?**

The upper size limit of the UV-APS and WIBS is nominally 20 um, though in practice the inlet design of an individual measurement site frequently lowers the upper size point somewhat.

**Comment**: 6. Page 10, lines 24-26: why are the Borneo and Nanjing data exceptional? The authors need to justify why they would remove these from the comparisons

**Response**: The Nanjing data reported too high concentrations that could be attributed partially to domestic pollution rather than biological particles (see the discussion related to these measurements). This has been added to the revised manuscript. For Borneo, we added it to the figure 6 with a different scale, and it has been included to the discussion.

**Comment**: 7. Section 3.5: the presentation of these results is a bit confusing. It would be helpful if the authors first discussed how many datasets are available for each season and commented on the observed seasonality before discussed the model performance.

**Response**: our point in this section is not to compare the seasonality of the model to observations. We believe that seasonality is not relevant here. The observations are only available for these specific time periods and our main objective is to compare the model concentrations to the available observations.

**Comment**: 8. Figure 6: It is very hard to see the data on this figure, suggest better use of scale (max value set to 0.1) to see data more clearly and using a color other than yellow/green which is hard to distinguish from white on the panels. The season labelling should be explained in the captions.

**Response**: We corrected the scale and changed the colors of the figure as the referee suggested.

**Comment**: 9. Page 11, line 1: unclear what "discrepancy" the authors are referring to. The measurements don't distinguish these two classes of PBAP.

**Response:** indeed the measurements don't distinguish between the two classes, but we expect from the measurements to see more fungal spores than bacteria (given all the uncertainties related to these measurements)

**Comment:** 10. Page 11, lines 25-28: clarify if these measurements are all for the same size ranges .

**Response**: Indeed, these measurements, as explained earlier in the paragraph and in Table 2, are for the same ranges.

**Comment:** 11. Page 12, line 19: reference of justification for assumed mass per particle needed

15   **Response:** The references have been added in the revised manuscript

**Comment**: 12. Page 12, line 20: specify that these means are for the surface

**Response**: The word "surface" has been added to the text.

**Comment**: 13. Page 12, line 28: The Poschl et al. numbers are averages over the deployment and only for supermicron particles and so cannot be directly compared to annual means of all aerosols from the model. Suggest that you compare to relevant month, coarse fraction only. If the large difference holds when the correct time of year is compared can the authors speculate as to why there would be such a substantial difference?

**Response**: We agree with the referee that this comparison is not appropriate therefore it has been removed.

**Response to Referee#2**

We thank the reviewer for the careful reading of the manuscript and helpful comments. We have revised the manuscript following the suggestions, as described below.

Summary

This manuscript has developed a global emissions and transport model for primary biological aerosol particles (PBAP). There have been several other prior attempts to do this for all types of PBAP (e.g., Jacobson and Street, 2009) as well as individual types (e.g., fungal spores Heald and Spracklen 2009; bacteria Burrows et al 2009a/b). In this regard, the work is

10   necessary but the manuscript itself does not make new advances in our simulations or understanding of global PBAP. In general, the manuscript has several major omissions of the data and methods driving the model, and this makes it not possible to interpret the results in any meaningful way. This paper requires major revisions to be acceptable in ACP.

Major comments

1. The manuscript states to have simulated the three main types of PBAP, yet there is no detail in the manuscript about the pollen emissions. The authors need to decide if are going to retain the pollen section of the PBAP inventory. If the authors choose to continue to include pollen in their total PBAP assessment, there are several new sec- tions that are absolutely necessary to understand what how the authors are simulating emissions and when and where they might be important. These

20   include the following:

**Comment**: a. Add a section 2.2.3 for pollen. Section 2.2 is titled "PBAP emissions," with one subsection devoted to bacteria (2.2.1) and the other to fungal spores (2.2.2), yet there is no corresponding section on the pollen emissions. A section explaining the pollen emissions parameterization is absolutely required. Similarly, this should include the size distribution

25   implemented in the model (as is for fungal spores and bacteria).

**Response**: We apologize for this omission in the text. The section has been added to the revised manuscript.

30   **Comment:** b. Provide some useful discussion on the pollen atmospheric distributions and their re- alism. For example, there is no discussion about the pollen emissions distributions that they simulate despite including a figure of pollen (Section 3.4). Specifically, their model simulates the highest concentrations of pollen in the tropics, which is inconsistent with the plant distribution of wind-driven pollen. Most plants in the tropics use insects or birds for transmission, so it is not expected that

there would be high emissions in these locations. It is impossible to determine why this is without the explanation of the pollen emissions model (see point above).

**Response**: The emissions depend on LAI similar to fungal spores, therefore, it should not be unexpected that concentrations are high in the tropics. It is difficult to discuss the realism of the global pollen distribution, as an important effort is needed to collect a global database for pollen. The Jacobson and Streets (2009) provided the first pollen parametrization to be used for a global model. This parameterization has not been evaluated against observations for any global model, and has been mostly used by the community to provide a global distribution for pollen, despite of its many deficiencies (e.g absence of the plant phenology and the land cover type). This was not the objective of the present study, as we focused on fungal spore distribution. In the discussion section, we recommend the use of a more recent parametrization developed and evaluated by Wozniak and Steiner (2017). For these reasons, we decided to remove Figures 5a and 5b from the main text and added them to the Supplementary Information, but we keep section 3.4, to be able to discuss the contribution of pollen to the total mass aerosol composition.

**Comment**: c. There are ground-based pollen count observations of similar spatial sparsity to fungal spores. Why is a comparison of measured versus modeled not included?

**Response**: Our purpose in this study is to focus on fungal spores parametrizations and to less extent bacteria. Pollen has been added to model as an additional PBAP only in order to estimate their contribution to the total aerosol mass (see section 3.6). A further modeling study using the new pollen emission paramerization proposed by Wozniak and Steiner (2017) will be the object of a following study including comparison with the available pollen count observations, especially in Europe, where a large pollen counts database is available.

**Comment:** d. In the final discussion, there is hardly any recommendations or future work regarding these emissions and improvements. Again, if the authors choose to keep pollen in there, a more rigorous discussion is required to explain the role of this specific type of biological particle.

**Response**: Recommendations have been added to the discussion section.

**Comment:** 2. There is no information about the LAI distribution used in the model. This is rather important for the fungal spore discussion, as much of the explanation for the three different fungal spores is often tied back to differences in leaf area.

**Response**: Both the HS and HU parametrization in our simulations use the same LAI distribution (as well as the pollen parametrization). Information about the LAI distribution used in our simulation has been added to the revised manuscript in section 2.1.

5 **Comment:** 3. Section 3.2: This type of correlation seems rather obvious: modeled LAI is of course going to be lower in urban areas, so then another factor would have to compensate (and it would be a meteorological parameter). I'm not sure how this would not be taken into account already by the existing models. If the authors think that this is important, then they should explore this in greater detail.

10 **Response**: We found these results interesting because they show the importance of the difference between urban and non-urban observations and their relationship with these physical parameters (usually not measured). This information could be taken into account potentially in future observations. Unfortunately, we could not find any observation publication reporting about these differences.

**Comment**: 4. Section 3.5: For the FBAP comparisons, why not also include the pollen and see if that improves any observations? The caveat of why FBAP may not (page 6, lines 20-21) or may (page 10, lines 10-13) is rather confusing.

**Response**: As shown in figure 5a, the pollen number concentrations are much lower that bacteria and fungal spores
20 concentrations, therefore their contribution to the total PBAP concentrations will be very low. Besides, as explained in Page 10, the upper size limit of the UV-APS and WIBS is nominally 20 um, though in practice the inlet design of an individual measurement site frequently lowers the upper size point somewhat, therefore pollen can not be included for comparison with FBAP observations.

Minor comments

**Comment**: 1. The title should reflect the full acronym of PBAP (e.g., add the term "aerosol")

30 **Response**: The word has been added to the title.

**Comment**: 2. Abstract Lines 26-27: Needed?

**Response**: We believe that these lines summarize one of the main findings of this paper, therefore they should be kept in the abstract.

**Comment**: 3. Page 2 Line 19: fungal spores reference about being the most abundant and genetically diverse – this is a rather old reference, is there any more modern support for this idea?

Response: The reference to the work of Fröhlich-Nowoisky et al. (2009) has been added

**Comment:** 4. Page 2 Line 35: Global and regional models are cited here, yet the papers primarily refer to global studies on fungal spores and bacteria. There is a wealth of literature out there on pollen, and this should be included if pollen is kept in the manuscript.

**Response**: Actually, Jacobson and Streets (2009), whose pollen parametrization has been used here and Hoose et al. (2010) include both pollen modeling.

**Comment**: 5. Page 5 line 31: Because the model was run without meteorological nudging, some brief reference to prior met evaluation of the model to indicate to readers that biases in the concentrations are not due to meteorological parameters such as temperature.

**Response**: References to the model evaluation has been added in section 2.1., as recommended by the referee.

Comment: 5. Page 4, section 2.2.1: More detail on the bacteria emissions, specifically the fact that they are constant and are not simulate with any meteorological dependency should be made very clear in this section. This is discussed later in the manuscript (e.g., Section 3.5) but it would be clearer to provide more detail here such that a reader is not looking up all the references.

Response: For the sake of clarity, we provide a table of the best-estimates fluxes in the Supplementary Material, and a reference to it in section 2.2.1.

**Comment:** 6. Page 4, line 29: what does "best estimate number fluxes" mean in this context?

**Response:** "best-estimate" is used considering the optimization method used by Burrows et al. (2009b) for the emission estimates to fit the observed number concentrations.

**Comment**: 7. Page 6, line 18: What are some examples of "highly fluorescent particles of non- biological origin," and would these be more likely to be observed in anthropogenically influenced areas?

**Response**: This sentence "such as certain kinds of aged brown SOA, diesel soot particles and some HULIS types" has been added to the text.

**Comment**: 8. Section 3.1: Note several references to Figure 1 that should be Figure 2, also, the fit metrics are not displayed as stated in the text (lines 12-13).

**Response**: This has been corrected in the revised manuscript.

**Comment**: 9. Section 3.1: Please clarify what model layers are used to compare to observed fungal spore counts, as this also may affect the model evaluation.

**Response**: The word "surface" has been added to the mean number concentrations.

[revised manuscript text omitted]

For the present study, we applied EMAC in the T63L31 resolution; with a spherical truncation of T63 (corresponding to a grid of approximately 1.9° x 1.9 ° in latitude and longitude, or approximately 140 **x** 210 km at middle latitudes), with 31 vertical hybrid pressure levels up to 10hPa. The model was run for five consecutive years without meteorological nudging from the year 2000 until 2004. AMIP-II monthly sea surface temperatures were used to provide boundary conditions for the atmospheric circulation, available for the period since satellite observations are available (1979). Climatological averages for PBAP distribution for the last four years of the simulation were used after a 1-year spin-up period. The EMAC model, evaluated in (Jöckel et al., 2005; Jöckel et al., 2016), and used in similar configurations, has been shown to be capable of realistic simulations of aerosol transport and deposition for the transport of African dust to Europe (Glaser et al., 2012) and radioactive aerosol particles from the Chernobyl accident (Lelieveld et al., 2012). We emphasize that the simulation results represent a climatology rather than specific weather conditions under which some PBAB samples may have been collected, hence we expect mean number concentrations and distributions to be represented by the model rather than distinct measurement data.

**2.2. PBAP emissions**

**2.2.1. Bacteria**

Bacterial emission fluxes are calculated using the constant best-estimate values from (Burrows et al., 2009a) for different ecosystems, which were optimized toward overall agreement with best-estimates of observation-based near-surface number concentrations (see Table SI1). We used the MODIS International Global Biosphere Program (IGBP) global land cover classification to determine the spatial distribution of 18 different ecosystems. We lumped the categories defined in the MODIS classifications to match similar sets of lumped ecosystems used by (Burrows et al., 2009a) (i.e., derived from the Olson ecosystem types), with the exception of the "urban" ecosystem, which is only present in MODIS data. We used a geometric mean diameter for bacteria of 4 μm for continental sources (forests, shrubs, grasslands, wetlands, savannahs and urban ecosystems) and 1.4 μm for marine sources. These choices are based on values reported for the count median diameter of bacteria-carrying particles, which may include bacteria borne by larger particles such as dust and leaf litter and/or clumps of bacteria (Shaffer and Lighthart, 1997; Tong and Lighthart, 2000, 1999; Wang et al., 2007). The export of particles simulated by the model varies as a function of both particle size and the geographic location of the emissions, which determines the atmospheric transport and removal processes the particles experience. However, the differences in atmospheric residence time and export that are associated with particle size are on the order of a factor of ca. 1.5 – 2.5 when varying the particle size between 1 um and 10 um (Burrows et al., 2013). The sensitivity of atmospheric transport to particle size for the EMAC model has been tested and described in detail in (Burrows et al., 2013; Kunkel et al., 2012). Therefore we note that modeled transport and removal processes are not strongly dependent on the particle size, at least not in the lower

µm size range, so that we do not consider the simplified size attribution of PBAPs to be a limiting factor in the representation of atmospheric processes.

**2.2.2. Fungal spores**

We compare three fungal spore emission parameterizations previously used in global and regional modeling studies. Firstly, fungal spore emission fluxes have been derived by (Heald and Spracklen, 2009) (HS hereafter) from an empirically optimized scheme where emissions are linear functions of the LAI (Leaf Area Index) and the specific humidity q at the surface. In order to match their emission estimates, (Hoose et al., 2010) applied the following formulation to calculate the emission flux in m$^{-2}$ s$^{-1}$, assuming a mean spore diameter of 5 µm:

$$F_{H\&D} = 500 m^{-2}s^{-1} \times \frac{LAI}{5} \times \frac{q}{1.5 \times 10^{-2} kg kg^{-1}}$$

Note that this formulation has been scaled to match the emission estimates by HS for a mean spore diameter of 5 µm.

The second parameterization we tested uses the emission number fluxes of fungal spores calculated by (Sesartic and Dallafior, 2011) (SD hereafter) for five different ecosystems (defined by (Olson et al., 2001)). We use the best-estimate number fluxes weighted by the area fraction of the respective MODIS ecosystems in the gridbox $E_i$. Note again that similar ecosystems from MODIS data are lumped according to the corresponding Olson ecosystems defined by (Sesartic and Dallafior, 2011). The total emission flux for fungal spores is given as in m$^{-2}$ s$^{-1}$:

$$F_{S\&D} = 194\ m^{-2}s^{-1} \times E_{tropicalforest} + 214\ m^{-2}s^{-1} \times E_{forest} + 1203\ m^{-2}s^{-1} \times E_{shrub} + 165\ m^{-2}s^{-1} \times E_{grassland}$$
$$+ 2509 \times m^{-2}s^{-1} \times E_{crop}$$

The third parameterization, derived by (Hummel et al., 2015) (HU hereafter), is adapted to measurements of airborne fluorescent biological particles across northern Europe. Similar to the parameterization of (Heald and Spracklen, 2009), this recent parameterization depends on LAI and specific humidity, and is extended to include temperature T:

$$F_{FBAP} = b_1 \times (T - 275.82) + b_2 \times q \times LAI$$

where $F_{FBAP}$ the emission flux in m$^{-2}$ s$^{-1}$, b1 = 20.426 and $b_2$ = 3.93 10$^4$, T is the surface temperature in K, q the specific humidity in kg kg$^{-1}$ and LAI the leaf area index in m$^2$m$^{-2}$

For each parameterization, the mean diameter was assigned according to the recommendation made for each in the three studies: 5 μm for HS, 3 μm for HU and 3 μm for SD. We reiterate that these size classifications are not expected to significantly influence the results.

**2.2.3. Pollen**

5     For pollen emissions, we use the emission parameterization proposed by (Jacobson and Streets, 2009) for use in global models. We apply it in a simplified form given by (Hoose et al., 2010), neglecting dependence on time of the day, relative humidity and turbulent kinetic energy:

$$F_{pollen} = 0.5 m^{-2} s^{-1} \times LAI \times R_{month}$$

$R_{month}$ is a factor accounting for seasonal variations (0.5 in the Northern Hemisphere for October to March, 2.0 for April to June, and 1.0 for July to September; and shifted by six months in the Southern Hemisphere). We assume a mean pollen

10     diameter of 20 μm. We stress the simplified nature of this parameterization. We have included it for consistency, and to obtain an approximation of the relative roles of different PBAP categories.

**2.3. Data description**

**2.3.1. Spores counts**

15     We compare the fungal spore concentrations calculated by EMAC using the three emission parameterizations described above, to observations collected through literature review. (Sesartic and Dallafior, 2011) have reviewed more than 150 studies and found that only a relatively small number, 35 of these, reported total fungal spore concentration measurements, excluding observations that employed cultivation of a subset of species (e.g., in a petri dish) and measurements that report only mass concentrations instead of spore counts. We updated this dataset with observations that meet the same criteria,

20     mostly from studies published since 2011. Our updated review revealed that much of the relevant literature reports only concentrations of the genetic diversity of fungal taxa and not their total concentrations, which explains the scarcity of data that can be used for model evaluation. The uncertainties related to these methods are discussed in detail in (Sesartic and Dallafior, 2011). The observations used for comparison with model results are listed in Table SI2. Modeled concentrations have been sampled from the output to match the period of observation for each location. Since we do not compute actual

[revised manuscript text omitted]

which the HS parameterization was based on, instead of the Olson distribution, which the SD parameterization was based on. The HU emission parameterization might not be suited for use in global modeling studies since it has been optimized for a regional modeling study over northern Europe. Differences in model physics, including the simulation distribution of precipitation, turbulent transport parameterizations, and parameterization of wet and dry removal, can also result in models simulating different concentrations, given the same emissions, so these results cannot necessarily be extended to other atmospheric models. In order to improve the original HS parametrization, an additional scaling factor of 6 has been added to the emission parametrization for our modelled concentrations to match the spore counts in Figure 2D. Although the correlation coefficient remains the same, this improves the MNMB and FGE metrics from 0.98 and 1.33 respectively to 0.18 and 1.00.

Discrepancies between model and observations may be explained by an over-prediction of fungal spore sources via biases in the emission parameterization as formulated by (Hoose et al., 2010) or long-range transport, or an under-prediction of the rate of removal by dry and wet deposition. Additionally, as outlined by (Sesartic and Dallafior, 2011) and references therein, the observational data quality is limited and should be considered with caution. The methods used to measure actual spore concentrations may involve biases as well as problems related to the identification of fungal spores. As mentioned in section 2.3.1, (Sesartic and Dallafior, 2011) showed that many direct-counting spore techniques can significantly undercount spore number (i.e. by order of magnitude). Additionally, any culture-based methods have significant biases in that only a very small fraction of spore species can be culturable in a given medium.

The total global emissions calculated here with HS (17 Tg yr$^{-1}$ corresponding to an average mass emission flux of 2.5 ng m$^{-2}$ s$^{-1}$) are within the range of uncertainties reported by (Després et al., 2012) and (Fröhlich-Nowoisky et al., 2016). Using the same HS parameterization, (Heald and Spracklen, 2009) and (Hoose et al., 2010) calculated higher totals, respectively 28 Tg yr$^{-1}$ and 31 Tg yr$^{-1}$. This demonstrates the model sensitivity to the leaf area index dataset used for that purpose and the specific humidity calculated by the model. The total global emissions calculated using SD and HU are estimated to 86 Tg yr$^{-1}$ and 349 Tg yr$^{-1}$, respectively, which seem unrealistically high. Further, the comparison of fungal spore number fluxes calculated by (Sesartic and Dallafior, 2011) and by EMAC yields large discrepancies in magnitude and spatial distribution. This is most likely explained by large differences in the biome spatial distribution between MODIS and Olson data, leading to the higher emissions calculated by EMAC when using the SD parameterization. Since the HS simulation shows a better fit to observations, we will show results only from this simulation hereafter.

[revised manuscript text omitted]

A

[Figure]

B

[Figure]

C

[Figure]

Figure 6: Near-surface annual mean chemical aerosol mass composition simulated by EMAC for five regions: North America (126°W – 72° W/30°N – 52° N), Europe (12°W – 36°E/34°N – 62°N), East Asia (100° - 144°E/20°N – 44°N), Central Africa (10°E – 40°E/10°S – 10°N) and South America (75°W – 35°W/30°S – 0°N). A. with dust, B. without dust, C. without dust and sea salt. The size of the pie chart is proportional to the total aerosol mass concentration.

**Table SI1.  Bacterial emission fluxes (best-fit) estimated by Burrows et al. (2009a).**

[revised manuscript text omitted]